# VDAC1 Silencing in Cancer Cells Leads to Metabolic Reprogramming That Modulates Tumor Microenvironment

**DOI:** 10.3390/cancers13112850

**Published:** 2021-06-07

**Authors:** Erez Zerbib, Tasleem Arif, Anna Shteinfer-Kuzmine, Vered Chalifa-Caspi, Varda Shoshan-Barmatz

**Affiliations:** 1Department of Life Sciences, Ben-Gurion University of the Negev, Beer-Sheva 84105, Israel; erezze@post.bgu.ac.il (E.Z.); tashu100@gmail.com (T.A.); 2Department of Cell, Developmental & Regenerative Biology, Icahn School of Medicine, Mount Sinai, NY 10029, USA; 3National Institute for Biotechnology in the Negev, Ben-Gurion University of the Negev, Beer-Sheva 84105, Israel; shteinfe@post.bgu.ac.il; 4Ilse Katz Institute for Nanoscale Science & Technology, Ben-Gurion University of the Negev, Beer-Sheva 84105, Israel; veredcc@bgu.ac.il

**Keywords:** metabolism, tumor microenvironment, mitochondria, reprogramming, siRNA, VDAC1

## Abstract

**Simple Summary:**

Tumors are comprised of proliferating cancer cells, and their microenvironment consists of the extracellular matrix, blood vessels, and a variety of tissue cells. The tumor microenvironment functions in cell growth, proliferation, migration, immunity, malignant transformation, and apoptosis. Understanding the molecular interactions between cancer cells and their microenvironment would facilitate the development of therapeutic strategies to disrupt these interactions and fight cancer. Here, we demonstrate that depleting the mitochondrial gatekeeper VDAC1 in human cancer cells in tumors led to metabolic reprogramming, inhibited tumor growth, and disrupted tumor–host interactions. A next-generation sequencing analysis of human lung cell-derived tumors expressing or depleted of VDAC1 allows distinguishing genes of human or murine origin, thus enabling the separation of the bidirectional cross-talk between malignant cells and the tumor microenvironment. A battery of human cancer cell and mouse genes associated with tumor microenvironment formation and remodeling were altered. The results point to VDAC1 as a novel target for both inhibiting tumor growth and modulating the tumor microenvironment, thus influencing cancer progression, migration, and invasion.

**Abstract:**

The tumor microenvironment (TME) plays an important role in cell growth, proliferation, migration, immunity, malignant transformation, and apoptosis. Thus, better insight into tumor–host interactions is required. Most of these processes involve the metabolic reprogramming of cells. Here, we focused on this reprogramming in cancerous cells and its effect on the TME. A major limitation in the study of tumor–host interactions is the difficulty in separating cancerous from non-cancerous signaling pathways within a tumor. Our strategy involved specifically silencing the expression of VDAC1 in the mitochondria of human-derived A549 lung cancer xenografts in mice, but not in the mouse-derived cells of the TME. Next-generation sequencing (NGS) analysis allows distinguishing the human or mouse origin of genes, thus enabling the separation of the bidirectional cross-talk between the TME and malignant cells. We demonstrate that depleting VDAC1 in cancer cells led to metabolic reprogramming, tumor regression, and the disruption of tumor–host interactions. This was reflected in the altered expression of a battery of genes associated with TME, including those involved in extracellular matrix organization and structure, matrix-related peptidases, angiogenesis, intercellular interacting proteins, integrins, and growth factors associated with stromal activities. We show that metabolic rewiring upon mitochondrial VDAC1 silencing in cancer cells affected several components of the TME, such as structural protein matrix metalloproteinases and Lox, and elicited a stromal response resembling the reaction to a foreign body in wound healing. As tumor progression requires a cooperative interplay between the host and cancer cells, and the ECM is intensively remodeled during cancer progression, VDAC1 depletion induced metabolic reprogramming that targeted both tumor cells and resulted in the alteration of the whole spectrum of TME-related genes, affecting the reciprocal feedback between ECM molecules, host cells, and cancer cells. Thus, VDAC1 depletion using si-VDAC1 represents therapeutic potential, inhibiting cancer cell proliferation and also inducing the modulation of TME components, which influences cancer progression, migration, and invasion.

## 1. Introduction

Cancer cells acquire metabolic adaptations in response to metabolic challenges during tumor progression, including growth in the nutrient-altered and oxygen-deficient microenvironment of the primary site. Indeed, cancer cells undergo metabolic reprogramming such as enhanced anaerobic glycolysis (Warburg effect) via upregulating the transcription of genes related to glycolysis [1,2]. However, mitochondria play a role in cancer cell-reprogrammed cellular metabolism [3], with metabolic flexibility serving to balance tumor cell energy needs with increased biosynthetic plasticity [2,4]. Growing evidence suggests that metabolism directly supports oncogenic signaling to sustain tumor cell growth [5,6]. However, tumors are complex tissues composed not only of cancer cells, but also of multiple distinct normal cells with their interactions creating the tumor microenvironment (TME), which actively participates in tumorigenesis. The TME includes fibroblasts [7] and immune system cells [8], as well as vasculature [9], the extracellular matrix (ECM) components [10], and other stromal factors that contribute to most of the properties of cancer [11]. Yet, the interplay between cancer cell metabolism and the host cells in the TME has not been well-studied.

The rigidity of the ECM was shown to affect fundamental cellular processes such as cell adhesion, growth, and migration [12,13]. These effects are particularly relevant for cancer since tumors are significantly stiffer than the tissues in which they grow due to increased deposition and crosslinking of ECM proteins at the tumor stroma. Indeed, high ECM stiffness was shown to affect several steps during the progression of malignant cancers, including promoting tumor growth and EMT [14,15]. The TME is a dynamic network of interactions between cancer cells, stromal tissue and the surrounding ECM involving fibrous proteins such as elastin, collagens, fibronectin, and laminins, as well as cellular proteases, such as cathepsins, matrix metalloproteinases, and heparinase, and proteins associated with intercellular interactions, integrins, and laminins [16]. 

As a major component of the TME, the ECM takes part in most basic cell behaviors, from cell proliferation, adhesion, and migration to cell differentiation, cell death, and oncogenic transformation [17]. The ECM is extremely versatile and performs many functions in addition to its structural role. Among the non-cancerous cell types commonly found in the TME, the most prominent are fibroblasts that function in the synthesis of many components of the ECM, such as collagens (type I–V) and fibronectin. Fibroblasts also participate in the regulation of epithelial cells via the secretion of growth factors and are involved in wound healing processes [18]. Moreover, fibroblasts secrete ECM-degrading proteases, such as matrix metalloproteinases (MMPs), highlighting the important roles of these cells in maintaining ECM homeostasis [19]. 

The ECM not only provides signals favoring migration, survival, and proliferation, but it also protects tumor cells from stress in terms of blood circulation or the immune response, and can influence drug resistance [20]. The evolution, structure, and activities of the cells in the TME share many parallels with the processes of wound healing and inflammation [1]. Fibroblasts play a critical role in wound repair, helping to generate ECM as a scaffold for other cells that participate in the wound healing process [19].

The TME crucially affects cancer progression and is a key regulator of metastasis [21], contributing to over 90% of cancer patient deaths [22]. In this multistep process of invasion, cancer cells leave the primary tumor and infiltrate nearby blood and lymphatic systems, followed by escape into the parenchyma of distant tissues (extravasation), the formation of micro-metastases and finally macroscopic tumors [1]. In addition, in order to leave the primary tumor, some cancer cells undergo epithelial–mesenchymal transition (EMT), namely the acquisition of a mesenchymal phenotype from a starting epithelial phenotype.

It has been shown that ECM proteins influence every step of the metastatic cascade of cancer. The ECM not only provides signals favoring migration, survival, and proliferation, but also protects the tumor cells from stress in blood circulation and immune response, and can influence drug resistance [20]. Many ECM proteins are associated with the induction of EMT by activating receptor-mediated signaling cascades, such as periostin [23], thus further strengthening the connection of the TME to cancer progression. Periostin is a secreted protein frequently overexpressed in many cancer types, with its expression being correlated with metastasis and poor patient survival [24,25]. Tenascin C (TNC), a secreted glycoprotein that binds to a variety of ECM proteins, such as periostin, fibronectin, integrins, and collagens [26], is overexpressed in several pathological conditions, including inflammation, wound healing, and cancer [27].

The link between TME and metabolism has been presented as several oncogenes and tumor suppressor signaling pathways regulate both cancer metabolism and modulate the cellular response to alterations in the TME during tumor progression [28,29]. In addition, conditions affecting the TME, such as hypoxia [30], acidosis [31], and tensile stress [10], exert critical influences on cancer cells. For example, intratumoral hypoxia causes cancer cells to rely on glycolysis for survival [32]. Aerobic glycolysis is a hallmark of advanced cancers [33], and is characterized by an increase in glucose uptake and lactate production, which diffuses and acidifies the TME [34]. This, in turn, alters the tumor–stroma interface, allowing for enhanced invasiveness [35] and proton release in the TME. 

Moreover, considering inflammation as a new hallmark of cancer [1,2], lactate provokes an inflammatory response that attracts immune cells, including macrophages that secrete cytokines, and growth factors that drive tumor cell growth, invasion, and metastasis [36]. Furthermore, lactic acid results in acidic TME stimulating angiogenesis and immune escape, disabling immune surveillance [37]. Notably, elevated serum lactate in cancer patients is associated with poor prognosis [38]. Thus, cancer cell metabolism controls not only energy supplies but also the TEM. 

The voltage-dependent anion channel 1 (VDAC1) is the mitochondrial protein that controls cell energy, metabolic homeostasis, and apoptosis [39,40] and is highly expressed in different tumors, including lung cancer, pointing to its significance in high energy-demanding cancer cells [40,41]. In mammals, three isoforms of VDAC (VDAC1, VDAC2, and VDAC3) have been identified, and they have been shown to share some, but not all, structural and functional properties. VDAC1 is the most abundant and best studied isoform and is highly expressed in different tumors, including lung cancer, pointing to its significance in high energy-demanding cancer cells [42]. Due to its localization at the outer mitochondrial membrane (OMM), VDAC1 is involved in a wide range of processes and interacts with many proteins [43,44]. VDAC1 acts as a channel, allowing the passage of ions, nucleotides, Ca^2+^, and other metabolites up to 5kDa, such as pyruvate, malate, succinate, NADH/NAD, and more, out and into the mitochondria, thereby regulating the function of the mitochondria. VDAC1 also plays a central role in apoptosis, mediating the release of pro-apoptotic proteins from mitochondrial intermembrane space to the cytosol via a large channel formed by its oligomerization, and interacting with anti-apoptotic regulators [39,40]. 

We have demonstrated that silencing VDAC1 expression in tumors by 2′-O-Methyl-modified siRNA specific to human (h)VDAC1 (si-hVDAC1-2A) reduced cellular ATP levels and cell proliferation, and inhibited solid tumor development and growth in glioblastoma, cervical, lung and breast cancers [41,45,46,47,48,49,50,51]. Moreover, using different methods such as immunofluorescence, immunohistochemistry, q-PCR, proteomics and next-generation sequencing (NGS) in several cancer types, we demonstrated that VDAC1 depletion in tumors resulted in major changes in the expression of metabolism-related enzymes such as glucose transport, glycolysis, the TCA cycle, electron transport, and ATP synthesis, and also affected epigenetic modifications involving mitochondrial metabolite-dependent processes [41,45,46,47,48,49,50,51,52]. As a key regulator of metabolic and energy reprogramming, disrupting cancer energy and metabolism homeostasis by VDAC1 depletion in tumor cells [53] is expected to also modulate the TME.

In this study, we addressed the relationship between tumor metabolism and the TME in lung cancer, a leading cause of cancer-related death worldwide [54]. Considering the heterogeneity and complexity of this deadly disease, a better understanding of the tumor–host interactions is required [55]. We focused on the interplay between metabolic reprogramming in the cancerous cells and of the TME. A major limitation in the study and of tumor–host interactions is the difficulty in separating cancerous from non-cancerous signaling pathways within the tumor. Accordingly, we performed a next-generation sequencing (NGS) analysis of human A549 cell-derived tumors in a mouse model treated with non-targeting siRNA (si-NT) or with siRNA specifically targeting human VDAC1 (si-hVDAC1-2A). As NGS analysis allows for distinguishing between genes of human and mouse origin, we were able to demonstrate the tumor–host interactions in lung cancer. We demonstrate that tumor cells affect and control the stromal, non-transformed elements of the TME. The results thus present the complex remodeling of the TME by cancer cells in a tumor, and suggest that VDAC1 depletion not only inhibits cancer cell proliferation, but that metabolic alterations in the cancer cells affect the tumor environment, influencing cancer progression, migration, and invasion. 

## 2. Materials and Methods

### 2.1. Materials

The materials used are summarized in Table 1.

siRNA specific to human VDAC1 modified with 2′-O-methyl at the indicated nucleotides highlighted in bold and underlined (si-hVDAC1-2A-2A) was synthesized by Genepharma (Suzhou, China). The sequences were: sense: 238-5′-ACACUAGGCACCGAGAUU A-3′-256 and antisense: 5′-UAAUCUCGGUGCCUAGUGU-3′. For a non-targeting siRNA (si-NT), the sequences were: sense: 5′-GCAAACAUCCCAGAGGUAU-3′ and anti-sense: 5′-AUA CCUCUGGGAUGUUUGC-3′.

### 2.2. Cell Culture and Transfection

A549 (human lung adenocarcinoma epithelial cell, non-small human lung carcinoma) and 2LL (mouse Lewis lung carcinoma) cells were obtained from ATCC and maintained at 37 °C and 5% CO_2_ in the recommended culture medium and supplements. Mycoplasma contamination in all cell lines was routinely tested using an EZ-PCR Mycoplasma Test Kit (Biological Industries, Beit Haemek, Israel). 

Cells were seeded (150,000 cells/well) in 6-well culture dishes and were transfected with si-NT or si-hVDAC1-2A at 40–60% confluence, using the JetPRIME transfection reagent, according to the manufacturer’s instructions. 

### 2.3. Mitochondrial Membrane Potential (ΔΨ) and Cellular ATP Levels

ΔΨ and cellular ATP levels were assessed as described in Text S1.

### 2.4. Xenograft Experiments

Lung cancer A549 (3 × 10^6^) cells were subcutaneously (s.c.) inoculated into the hind leg flanks of 7-week-old Nude-Foxn1nu Hsd: athymic male nude mice (Envigo, Jerusalem, Israel). When the tumor volume reached 60–70 mm^3^, the mice were divided into two groups: one group was intratumorally treated with si-hVDAC1-2A mixed with the in vivo JetPEI reagent to a final concentration of 200 nM (calculated according to the tumor volume, 2 boluses) every 3 days. When the mice were sacrificed, the tumors were excised and processed for immunohistochemistry, or frozen in liquid nitrogen for immunoblotting and RNA isolation, as described in Text S1. Experimental protocols used were approved by the Institutional Animal Care and Use Committee.

### 2.5. Sirius Red Staining

Sirius red staining for collagen fibers was carried out as described previously [56]. Briefly, tumor tissues were fixed and embedded in paraffin sections, and were stained with a 0.1% Sirius redsolution. Sections were washed rapidly with acetic acid and photographed under a light microscope (Leica DM2500).

### 2.6. Immunohistochemistry (IHC) and Immunofluorescence (IF) Staining of Tumor Tissue Sections

IHC and IF staining were performed on 5 μm-thick formalin-fixed and paraffin-embedded tumor tissue sections as described previously [49]. Tumor sections following deparaffinization were incubated overnight with primary antibodies and then for 2 h with HRP-conjugated secondary antibodies, as appropriate (Appendix A). Sections were incubated with DAB, the HRP substrate, counterstained with hematoxylin, observed under a microscope (Leica DM2500), and images were collected at the same light intensity and exposure time. For immunofluorescence, fluorophore-conjugated secondary antibodies were used (Appendix A), nuclei were stained with DAPI (0.07 μg/mL) and sections were viewed with an Olympus IX81 confocal microscope.

Quantitation of protein levels, as reflected in the staining intensity, was analyzed in the whole area of the sections using Image J software (version 1.43 u).

### 2.7. Tumor Tissue Extracts and Immunoblotting

Tumor tissues were lysed using lysis buffer (50 mM Tris-HCl, pH 7.5, 150 mM NaCl, 1 mM EDTA, 1.5 mM MgCl_2_, 10% glycerol, 1% Triton X-100), supplemented with a protease inhibitor cocktail (Sigma, St. Louis, MO, USA). Following centrifugation at 12,000× *g* (10 min at 4 °C), aliquots (10–40 μg of protein) were subjected to SDS-PAGE and immunostaining using the selected primary antibody (Appendix A) and the appropriate secondary HRP-conjugated anti-mouse or anti-rabbit antibody (Appendix A). Enhanced chemiluminescent substrate EZ-ECL (Biological Industries, Kibbutz Beit-Haemek, Israel) was used for detection of HRP activity. Band intensity was quantified using FUSION-FX (Vilber Lourmat, Marne La Vallee, France).

### 2.8. RNA Preparation and Quantitative RT-PCR (q-RT-PCR)

Total RNA was isolated from si-NT- and from si-hVDAC1-2A-treated tumors (TTs) (from three mice each) using an RNeasy mini kit (Qiagen; Hilden, Germany). Complementary DNA was synthesized from 1 µg total RNA using a Verso cDNA synthesis kit (Thermo Scientific; Walthman, MA, USA). q-RT-PCR was performed as described previously [49], using specific primers (Appendix A, Sigma Aldrich (St. Louis, MO, USA)) in triplicate, with Power SYBER Green Master Mix (Applied Biosystems; Foster City, CA, USA). β-actin mRNA was used to normalize the levels of the target genes. The results are presented as mean fold change (±SEM) of the three replicates.

### 2.9. NGS and Bioinformatics Analyses

Libraries were prepared from double-stranded cDNA produced from mRNA that was isolated from the tissue samples by the INCPM (Nancy and Stephen Grand Israel National Center for Personalized Medicine, Rehovot, Israel) RNA-seq unit (Weizmann Institute of Science, Rehovot, Israel) as described previously [57]. Libraries were evaluated by Qubit and TapeStation. Sequencing libraries were constructed with barcodes to allow multiplexing of 20 samples per lane. Between 22 and 26 million single-end 60-bp reads were sequenced per sample on an Illumina HiSeq 2500 V4 instrument.

Bioinformatics analyses were carried out at the Bioinformatics Core Facility at the Ben-Gurion University (Beer-Sheva, Israel) as described previously [57]. Human- and murine-specific, uniquely mapped genes were analyzed. Differentially expressed genes were defined as those having a *p*-value < 0.05, and a linear fold change >1.5 and <−1.5. Functional analysis was performed using the Gene Ontology system, DAVID, and Expander software tools. 

Functional classification of differentially expressed genes to GO-Slim Biological Processes was performed using Panther [58].

### 2.10. Statistics

Data are expressed as mean ± standard error of the mean (SEM) of results from independent experiments. Unpaired two-tailed Student’s t-test was used for all experiments. A difference was taken as statistically significant when the *p*-value was ≤0.05 (*), ≤0.01 (**), ≤0.001 (***), ≤0.0001 (****). 

## 3. Results

Growing evidence suggests that metabolism directly supports oncogenic signaling, with cancer progression that is correlated with a high metabolism rate [59]. In our previous studies, we demonstrated, both in vitro and in mouse xenograft models of human glioblastoma (U-87MG), lung cancer (A549), and triple negative breast cancer (MDA-MB-231), that silencing VDAC1 expression using human-specific siRNA (si-hVDAC1-2A) inhibited cancer cell growth [41,46,47,48,49,50]. Moreover, silencing VDAC1 expression in the tumors resulted in multiple effects, including highly reduced expression of metabolism related enzymes, eliminating tumors’ oncogenic properties (e.g., angiogenesis, stemness), and inducing differentiation into normal-like [41,46,47,48,49,50].

Here, we focused on another important tumor property, the cancer cell–TME and ECM interactions in lung cancer tumors, demonstrating how tumor cell metabolism impinges on the microenvironment and ECM.

Our approach was to specifically silence the expression of mitochondrial hVDAC1 in human-derived A549 lung cancer xenografts in mice and then perform NGS analysis, allowing genes of human and mouse origin to be distinguished, thereby demonstrating that cancer cell metabolism is a key factor controlling both the cancer cell and its microenvironment within the tumor.

### 3.1. si-hVDAC1-2A Inhibits Tumor Growth and Reduces Energy Production in a Lung Cancer Xenograft Model

The VDAC1 sequence is greatly conserved between humans and mice, differing in only 3 of the 282 amino acids comprising the protein. The 21 bp human VDAC1-siRNA (hVDAC1-2A) used in this study thus corresponds to a region of the hVDAC1 nucleotide. The specificity of the siRNA against human VDAC1 was demonstrated previously for human HCT-116 and murine CT26 colon carcinoma cells [41]. Here, we demonstrated its specificity using human A549- and mouse 2LL-derived lung cancer cell lines (Figure 1A,B). The results show that si-hVDAC1-2A prevented VDAC1 expression in cells of human origin, but not in murine cells.

Silencing VDAC1 expression by si-hVDAC1 in A549 cells leading to marked decreases in VDAC1 levels (Appendix A), as expected [41,45,46,47,48,49,50,51], highly decreased the mitochondrial membrane potential (ΔΨ) and cellular ATP levels (Appendix A), resulting in cell growth inhibition (Appendix A). si-NT had no significant effect on the analyzed parameters (Appendix A).

The finding that si-hVDAC1-2A specifically targets human VDAC1 mRNA allows its effect to be on human A549 lung cancer cell-derived xenografts, but this does not affect the mouse host cells within the tumor. Thus, next, we treated an established A549 lung cancer cell-derived xenograft in a nude mice model with non-targeting siRNA (si-NT) and si-hVDAC1-2A. When the tumor volume reached (~70 mm^3^), the mice were split into two matched groups, with the tumors being injected every 3 days with si-NT or si-hVDAC1-2A. The volume of the si-NT-TTs increased from 72 mm^3^ on day one of treatment (day 11 post-inoculation) to 201 mm^3^ on day 22 of treatment (day 33 post-inoculation), corresponding to an average increase in tumor volume of 279% (Figure 1C,D). In contrast, the tumor growth of the si-hVDAC1-2A-TTs decreased with each sequential treatment from a tumor volume of 68.3 mm^3^ on day one of treatment to 37 mm^3^ on day 22 of treatment, corresponding to an average decrease in the starting tumor volume of 45% (Figure 1D). Thus, si-VDAC1 treatment not only stopped tumor growth, but induced its regression to a size 18% of that of si-NT-TTs.

Next, VDAC1 expression levels in the tumors were analyzed using immunoblotting and IHC (Figure 1F,G). Tumor tissue lysates were immunoblotted with anti-VDAC1 antibodies recognizing both human and mouse VDAC1. VDAC1 levels in the si-hVDAC1-2A-TTs were decreased by about 80% compared to those in si-NT-TTs, even though the residual VDAC1 may have originated in the mouse cells (Figure 1E). IHC staining of tumor sections was performed to assess the expression levels of metabolism-related enzymes such as the glucose transporter (GLUT1), glyceraldehyde 3-phosphate dehydrogenase (GAPDH), TCA-cycle enzyme citrate synthase (CS), mitochondrial electron transport complex IVc (Comp. IVc), and ATP synthase subunit 5a (ATPsyn 5a). The levels of all these enzymes at the protein levels were reduced 55 to 95% in the si-hVDAC1-2A-TTs, and at mRNA were reduced 2 to 13-fold, relative to their levels in si-NT-TTs (Figure 1F–H). This is consistent with alterations in glycolysis and oxidative phosphorylation (OXPHOS) metabolic pathways and in agreement with the concept that cancer cells use a combination of glycolysis and mitochondria to produce energy [60,61].

### 3.2. Next-Generation Sequencing (NGS) of si-NT- and si-hVDAC1-2A-TTs Reveals Changes in the Expression of Mouse ECM Structure-Related Genes upon si-hVDAC1-2A Treatment 

To explore changes in the patterns of mouse gene expression in tumor tissue from lung cancer xenografts derived from the A549 human cell line upon si-hVDAC1-2A-TTs treatment, we performed next-generation sequencing (NGS) analysis of si-NT-TTs and si-hVDAC1-2A-TTs, summarized in the figures below and Appendix A. This analysis revealed a total of 3134 genes that displayed significant differential expression (≥1.5-fold change, *p-*value < 0.05) between si-NT- and si-hVDAC1-2A-TTs. Of these, 1537 (49%) were of human origin and 1597 (51%) were of murine origin. Of the human genes whose expression was modified following hVDAC1 silencing, 1288 were upregulated and 249 were downregulated. As for the mouse genes, 901 were upregulated and 696 were downregulated (Figure 2A,B).

Panther functional classification of the differently expressed human (cancer cells) and mouse (TME) genes in the si-hVDAC1-2A-treated tumors revealed alterations in key functions and pathways (Figure 2C–F). The major functional groups with altered expression included the cellular processes-related genes, with 809 and 513 genes for mice and humans, respectively, and biological regulation-related genes with 471 and 323 genes, and metabolic processes-related genes with 441 and 284 genes for mice and humans, respectively. Other interesting groups of genes that were differentially expressed between si-hVDAC1-2A-TTs and si-NT-TTs include signaling (206 mouse, 142 human) and response to stimuli (281 mouse, 186 human). It should be noted that some genes were mapped to more than one biological process. As the focus of this study was on the metabolism-related genes, we present the various processes within this group (Figure 2E,F). The results show alterations in the expression of key pathways related to metabolism both in those of human origin derived from human cancer cells (A549 cells) where VDAC1 was silenced and in those of mouse origin, representing the TME.

The ability to distinguish between differentially expressed genes of mouse and human origin, in combination with the human-specific siRNA used in the xenograft model, allowed us to investigate changes that occurred in the TME due to hVDAC1 depletion in the cancer cells. One of the most important facets of the TME that greatly affects tumor progression is the dynamics of the ECM, which constantly undergoes a remodeling process, whereby ECM components are deposited, degraded, or otherwise modified. The expression levels of ECM structure-related genes of murine origin, including collagens, glycoproteins, integrins, and EMC-organizing genes, were found to be significantly reduced in the si-hVDAC1-2A-TTs (Figure 3, Appendix A). Collagen genes (COL1A1, 1A2, 2A1, 3A1, 4A1, 5A1, 6A1, 8A1, 8A2, 11A1) were significantly downregulated 2- to 4.5-fold (Figure 2A, Appendix A). Fibrillar collagens contribute to the major tensile strength of the ECM [62]. Of the glycoprotein components of the ECM, the expression of several gene families was reduced (2–4.5-fold), among them the laminins (LAMA2, LAMA4, LAMB1, LAMB2, LAMC1), fibulins (FBLN1, FBLN2), fibrillins (FBN1, FBN2), fibronectin (FN1), and nidogens (NID1, NID2) (Figure 3B, Appendix A). The proteins encoded by these genes participate in building the matrix network, serving as connector or linking proteins [62].

Additionally, the expression levels of heparan-sulphate-associated genes were also decreased (HS3ST1, HS2ST1, HS3ST1) 2.3 to 1.7-fold (Appendix A). These results indicate that upon hVDAC1 depletion, the expression of genes encoding structural components of the ECM was reduced, pointing to modified activity of the fibroblast population residing in the TME.

### 3.3. si-hVDAC1-2A Alters ECM Organization-Related Genes 

The expression levels of some mouse integrins in the si-VDAC1-2A-TTs (ITGBL1, ITGB5, and ITGAV) decreased (1.5–4.5-fold), while others increased (ICAM1, ITGB2, ITGB4, ITGA2B, and ITGAX) (2.5–3.5-fold) (Figure 3C, Appendix A). Integrins are the major adhesion receptors of the ECM, controlling ECM assembly, cell–matrix interactions, cell migration, and tumor growth [63]. They bind extracellular matrix fibrils and associate with intracellular actin filaments through a variety of cytoskeletal linker proteins to connect intracellular and extracellular structures. Thus, integrins are key mediators of both cell-to-matrix and cell-to-cell adhesive interactions. As these proteins are associated with the attachment and migration of cells, the changes in their expression levels indicate that a different organization of the TME would affect metastasis formation. 

A group of genes associated with organization and functions of ECMs was found to be significantly reduced (2 to 4-fold) in the si-hVDAC1-2A-TTs (Figure 3D, Appendix A), and among them are the two key modulators of the TME—periostin (also called POSTN) and tenascin C (TNC) (Appendix A). Due to their importance and multi-functions with respect to the ECM, we further analyzed the effect of tumor treatment with si-hVDAC1-2A on the expression of periostin and tenascin C. 

Periostin is a secreted extracellular matrix protein that plays an important role in proper collagen assembly and homeostatic functions in tissue development and regeneration, including wound healing, and it is a ligand for integrins [64,65]. Tenascin C is a multi-functional ECM glycoprotein able to regulate cell behavior and matrix organization while remodeling, contributing to the formation of both reactive and replacement fibrosis [66], and it promotes epithelial–mesenchymal transition (EMT), proliferation, and migration of cancer cells [67]. The expression levels of periostin and tenascin C were analyzed by IF of sections from si-hVDAC1-2A-TTs and si-NT-TTs to evaluate protein levels in the tumor. The si-NT-TT sections were more strongly stained for both periostin and tenascin C in the ECM region, as compared to the staining seen in si-hVDAC1-2A-TTs (Figure 4A–D). The staining in the tumor–stroma interface appeared as a network consisting of aligned fibers (Figure 4C).

si-hVDAC1-2A tumor treatment altered the expression of other genes associated with ECM organization (Appendix A). These results demonstrated that VDAC1 depletion leading to metabolic reprogramming affects a large spectrum of proteins associated with ECM organization.

### 3.4. NGS Analysis of si-NT-TTs and si-hVDAC1-2A-TTs Reveals Reduced Expression of ECM Deposition- and Degradation-Related Genes

ECM dynamics involving not only protein production but also their degradation were affected upon the depletion of VDAC1 in cancer cells (Figure 5 and Appendix A). ECM degradation is largely attributed to the activity and expression levels of matrix metalloproteinases (MMPs). NGS analysis showed a significant decrease in MMP expression levels in the si-hVDAC1-2A-TTs (Figure 5A, Appendix A). Levels of MMP 2, 3, 9, 11, 13, 14, 23, and 27 were reduced 2.5–4-fold, while the expression of MMP12 was markedly increased (~5-fold, Appendix A). Interestingly, in the murine models, MMP12 protects against tumor progression [68,69] activity, which has been ascribed to the generation of anti-angiogenic peptides due to MMP12 activity [70].

Other families of proteinases, a disintegrin, and metalloproteinases (ADAMs) involved in both proteolysis and cell adhesion (ADAM 9, 12, 22) and a disintegrin and metalloproteinase with thrombospondin motifs (ADAMTSs) (ADAMTS1, 2, 4, 5, 7, 9, 12, 13, 15, and 16) showed 2- to 8-fold decreased expression in si-hVDAC1-2A-TTs relative to their expression in si-NT-TTs (Figure 5B, Appendix A). 

The expression levels of the lysyl oxidase gene family (*loxl 1*, *2*, and *4*), responsible for the formation of crosslinks in collagens and elastin, were decreased (1.7–3.5-fold) (Figure 5C, Appendix A). LOX was found to be overexpressed in lung cancer and in many other cancer types, and it is correlated with tumor metastasis [71]. High LOX activity has been correlated with ECM stiffness and poor prognosis in breast, head and neck, colorectal, and prostate cancer [72]. Targeting LOX by inhibition was shown to decrease lung cancer progression [73].

In addition, the cell surface glycoprotein prolyl endopeptidase FAP (FAP), which participates in ECM degradation and is involved in many cellular processes including tissue remodeling, fibrosis, wound healing, inflammation and tumor growth, was decreased about 4-fold (Figure 5D, Appendix A)

Another group of proteins whose expression levels were highly reduced (2 to 10-fold) are associated with cell–ECM interactions and effectors of the stromal tissue (Figure 5D, Appendix A). These proteins include tenascin-N (TNN), overexpressed in most mammary tumors and associated with migration [74]; integrin-binding sialoprotein (IBSP), which protects cells from complement-mediated cellular lysis [75]; and transforming growth factor-beta (TGF-b 2 and 3), which plays an important role in mediating fibrotic tissue remodeling [76]. The decreased expression of the TGF-β family members and FAP can explain the change in expression of some of the proteins indicated above.

Together, these findings point to low levels of ECM remodeling and reduced fibroblast activity, resulting in deregulated cell proliferation and invasion.

### 3.5. hVDAC1 Depletion in a Tumor Changes ECM Organization and Collagen Levels by Altering the Ability of Activated Fibroblasts to Encapsulate Cancer Cells in the Tumor

To further evaluate the tumor morphology following si-hVDAC1-2A-induced alteration in the expression of mouse genes associated with TME, si-hVDAC1-2A-TT and si-NT-TT sections were stained for collagens with Sirius red. A stronger stain was observed in the si-hVDAC1-2A-TTs (Figure 6A,B). Sirius red staining revealed changes in fibrosis morphology in si-hVDAC1-2A-TTs, where collagen fibers were found to encapsulate large portions of the tumor. Spherical collagen structures surrounding tumor cells were found throughout these sections (marked by dotted lines, Figure 6A), as compared to the distracted and sporadic fibrils found in si-NT-TTs. Since fibroblasts are the main source of collagen fiber formation, we next IHC- and IF-stained for the fibroblast marker alpha smooth muscle actin (α-SMA) using specific antibodies. α-SMA staining showed a similar structural organization as was obtained with Sirius red staining, where fibroblasts were found to encapsulate tumor cells in si-hVDAC1-2A-TTs, whereas the staining in si-NT-TTs pointed to random fibroblast localization (Figure 6C,D). α-SMA staining intensity was increased by 2-fold in si-hVDAC1-2A-TTs relative to si-NT-TTs (Figure 6E). These findings point to a possible correlation between collagen encapsulation and fibroblast activity that highly resembles a wound healing process.

### 3.6. Tumor Treatment with si-hVDAC1-2A Altered the Expression of Mouse Genes Associated with Angiogenesis of TME Cells

Angiogenesis is an underlying promoter of tumor growth, invasion, and metastases. The expression of many genes can affect this process in a variety of ways, with some having ambivalent effects, depending on the tissue context. RNAseq data analysis revealed that tumors treated with si-hVDAC1-2A show altered expression profiles of a significant number of angiogenesis-associated genes (Figure 7, Appendix A). One of the prominent families of pro-angiogenesis factors that showed substantial decreased expression in the si-hVDAC1-2A-TTs was the angiopoietins (ANGPTs). These are vascular growth factors that play a role in embryonic and post-natal angiogenesis and control microvascular permeability, vasodilation, and vasoconstriction by signaling smooth muscle cells surrounding vessels.

Three of the four family members, ANGPT1, ANGPT2, and ANGPT4, were found to be reduced 2- to 5-fold (Figure 7A, Appendix A). Another group of proteins showing reduced expression included platelet-derived growth factor receptors (PDGF-Rs), cell surface receptors with intracellular tyrosine kinase activity, which regulate many biological processes including embryonic development, angiogenesis, cell proliferation, and differentiation [77]. These receptors, which bind platelet-derived growth factor (PDGF), with subunits -A and -B being factors that regulate cell proliferation, cellular differentiation, cell growth, development, and many diseases including cancer, were decreased 2–4-fold in si-hVDAC1-2A-TTs (Figure 7A, Appendix A).

In contrast, increased expression of transforming growth factor alpha (TGF-α) (3-fold); prospero homeobox protein 1 (PROX1) (8-fold); castor zinc finger 1 (CASZ1) (12-fold), a transcription factor involved in vascular assembly; and ectonucleotide pyrophosphatase/phosphodiesterase family member 2 (ENPP2) (16-fold), also known as autotaxin, which contributes to the metastatic cascade via promoting angiogenesis [78], was seen in si-hVDAC1-2A-TTs (Figure 7A, Appendix A).

The expression of several genes with anti-angiogenic activity was increased (Figure 7B, Appendix A). These include C-X-C motif chemokine 10 (CXCL10) (10-fold), a chemokine associated with the inhibition of endothelial cells and the activity of the immune system [79]; interferon gamma (IFNγ) (12-fold), a cytokine with immunostimulatory and immunomodulatory effects [80]; and interleukin 12 (IL12) (26-fold), a multi-functional cytokine associated with the inhibition of angiogenesis (Figure 7B,C,E, Appendix A). 

The effects of silencing VDAC1 expression on angiogenesis are reflected in the decreased immunostaining of endothelial cell marker anti-CD-31 (Figure 7C) with decreased staining intensity by about 80% (Figure 7D).

### 3.7. si-hVDAC1-2A Effects on the Expression of TME-Related Human Genes 

Finally, as the tumor contains A549 human cell-derived proteins, we analyzed the effect of si-hVDAC1-2A treatment on TME-related genes (Figure 8A–D, Appendix A). Decreased expression levels of proteins associated with ECM-related structural genes, such as COL8A2, COL1A1 and COL4A1 of about 4.5-, 2-, and 1.9-fold, respectively, were obtained (Figure 8A, Appendix A). The expression of several genes with anti-angiogenic activity such as C-X-C motif chemokine 10 (CXCL10) (6-fold), a chemokine associated with the inhibition of endothelial cells and the activity of the immune system [81], was altered (Figure 8B). On the other hand, some factors associated with pro-angiogenic properties showed decreased expression after treatment with si-hVDAC1-2A, such as PDGFC, PDGFRB, and APLNR (Figure 8B and Appendix A). As expected, the expression of anti-angiogenesis-associated proteins such as ENPP2 and filamin A-interacting protein 1-like (FILP1L) showed 16.7 and 11-fold increased expression, respectively (Appendix A).

Such analysis also revealed alterations in ECM organization proteins, such as cell migration-inducing and hyaluronan-binding protein (CEMIP) and latent-transforming growth factor β-binding protein 2 (LTBP2), that were decreased about 4- and 1.8-fold, respectively (Appendix A). ECM proteases ADAMTS12, MMP13, and MMP2 showed 3.9-, 2.5-, and 2.1-fold decreased expression, respectively (Figure 8C, Appendix A). Intercellular interaction proteins such as endomucin (EMCN), ITGA11 and integrin-binding sialoprotein (IBSP) were 8-, 2.6- and 10.7-fold less expressed, respectively (Figure 8D, Appendix A), while growth factors such as TGFb2 and insulin-like growth factor 2 (IGFL2) were 2 and 4.2-fold less expressed, respectively (Figure 7D, Appendix A). 

These results indicate that the reprogramming of cancer cell metabolism by VDAC1 depletion modulates both TME-associated gene expression in the host cells and the cancer cells, supporting bidirectional cross-talk between the microenvironment and malignant cells.

### 3.8. Validation of the Effects of si-hVDAC1-2A on Key Mouse TME Factors by q-RT-PCR

To validate the results obtained with NGS analysis, we performed q-RT-PCR on selected genes whose expression levels changed following tumor treatment with si-hVDAC1-2A. Remarkably, although another A549-derived xenograft experiment treated with si-hVDAC1-2A (Figure 9A–C) was used for this validation, the q-RT-PCR results for selected key TME-related genes showed a similar increase or decrease in expression levels as found in the NGS results. VDAC1 expression levels in the tumors were analyzed using immunoblotting (Figure 9C). Tumor tissue lysates were immunoblotted with anti-VDAC1 antibodies recognizing both human and mouse VDAC1. VDAC1 levels in the si-hVDAC1-2A-TTs were decreased by about 80% when analyzed by immunoblotting (Figure 9B) and by 60% when analyzed by IF (Figure 9D,E).

For most genes, the fold of change was similar in both NGS and q-RT-PCR analyses, except for MMP2 and PLOD2, which showed decreased expression of about 4.5- and 3-fold, respectively, as compared to 17- and 10-fold in the NGS results (Figure 9F). As such, these results further support the effects obtained upon depleting VDAC1 in tumors on host cells, as represented in the altered expression of TME-related genes.

## 4. Discussion

In recent years, the microenvironment surrounding tumor cells has been considered to be one of the main players in cancer development and strongly influences tumor onset and progression. The cancer microenvironment is associated with smoldering inflammation induced by tumor-associated macrophages, hypoxia, low pH, and the mobilization of immunosuppressor cells, such as immature myeloid cells, Tregs cells, and stromal activity. All of these may attenuate anticancer immune system activity and the escape of cancer cells from immune system control [82].

This type of immune escape is related to decreased expression of ICIs such as PD-L [83], resulting in treatment insensitivity [84]. Clinical experience has shown that only 15–25% of patients with various types of cancer respond to anti-CTLA-4 or anti-PD-1/PD-L1, and the treatment prolongs survival by only 3.4 months [85].

In this study, we focused on the effects of metabolic reprogramming via depleting specifically VDAC1 in the human cancer cell on the TME derived from a host mouse. The ability to manipulate the TME to its advantage is a defining feature of cancer, especially in the remodeling of the ECM, which helps to maintain and support cancer proliferation and progression [86]. However, the specific mechanisms by which cancer cells perturb the TME to influence the development, progression, and response to therapy are not fully understood. 

Cross-talk between the host and non-cancerous and cancer cells within a tumor is bidirectional. A major limitation in the study of tumor–host interactions is the difficulty in separating cancerous from non-cancerous signaling pathways and components within a tumor. To overcome this difficulty, we used small interfering RNA specifically targeting the mRNA of human VDAC1 that does not recognize mouse (m)VDAC1 (Figure 1B). Using A549 lung cancer cells of human origin in a mouse model together with NGS analysis enabled advanced molecular analysis. This approach further allowed for distinguishing between human and mouse mRNA variants, and thus provided a glimpse into the microenvironment to identify the signaling pathways that are activated in the host cells by cancer cells. Furthermore, we demonstrated how the reprogramming of energy and metabolism in cancer cells affects different components of the TME. Understanding the mechanisms of ECM remodeling and regulation by cancerous cells can contribute to developing new therapeutic interventions via TME removal or hindrance. Here, we presented novel strategies of metabolic reprogramming designed to alter the TME.

### 4.1. The Link between Reprogramed Cancer Cell Metabolism and the TME

In our previous studies, the effect of VDAC1 depletion in glioblastoma (GBM), lung, and triple negative breast cancer on tumor oncogenic properties was analyzed in the context of reprogramming metabolism, invasion, proliferation, stemness, differentiation, and angiogenesis modulation, but not on TME components [48,49]. In this study, we used human lung adenocarcinoma in a murine xenograft model to study interactions between cancer and the microenvironment. 

The link between metabolism and the TME in cancer plasticity has been addressed [87], with bidirectional cross-talk being shown. The biochemical features of the TME, such as hypoxia, acidosis, and nutrient deprivation, significantly regulate cancer cell metabolism, and reciprocally, de-regulated cancer cell metabolism leads to an accumulation of metabolic by-products such as lactate and protons in a tumor, which substantially modulate the TME [88]. Here, by silencing the expression of VDAC1 specifically in cancer cells, we were able to modify their metabolism (Figure 1 and Figure 2) and view the effect on the microenvironment. VDAC1, as the mitochondrial gatekeeper and overexpressed in cancers [89], is critical for cancer progression, with its depletion inducing metabolic reprogramming and arresting tumor growth [48]. In lung cancer, this even leads to the regression of an established tumor (Figure 1). 

Interestingly, the residual “tumor” in the si-hVDAC1-2A-TTs showed a notably different morphology and cell arrangement, as visualized by Sirius red and a-SMA staining (Figure 5). The organization of fibroblasts observed in the TME was remarkably similar to that usually seen in granulation tissues of healing wounds [90]. Fibroblasts were identified surrounding cancer cells, and collagen fibers were deposited to encapsulate a portion of the tumor. This phenomenon highly resembles the end-stage response of wound healing and foreign body reaction processes and may be indicative of more benign behavior of the tumor. A change in the mouse integrin expression profile further supports this similarity, pointing to a tendency that is more adhesive than migratory (Figure 3C). 

In addition, the expression of periostin, a pro-fibrogenic secreted glycoprotein, is defined as a matricellular protein based on its expression pattern and regulatory roles during development and healing and in disease processes [91]. The expression of periostin, a member of a class of non-structural ECM proteins that modulate cell–matrix interactions and plays an important role in proper collagen assembly and homeostasis, tissue development and regeneration, including wound healing [91], was decreased (Figure 4A). Periostin interacts directly with collagen type I and fibronectin, components of fibrillar networks, and tenascin-C, all of which were also reduced in si-hVDAC1-2A-TTs (Figure 4, Appendix A), and it organizes the extracellular meshwork, with its formation associated with the aggressiveness of metastasis. Indeed, periostin, collagen type I, and fibronectin were shown to be upregulated in a metastatic lesion that developed at a wound site in a patient with melanoma [92]. Additionally, periostin was shown to promote cell motility in ovarian cancer [65], while anti-periostin antibodies inhibited breast cancer progression and metastasis [25]. Similarly, antibodies targeting tenascin C delayed tumor growth and induced apparent cures in murine xenograft models [93].

As one of the important aspects of the TME, angiogenesis is often enhanced during tumor development [94]. This aspect is shared with the wound healing process since new blood vessel formation is crucial for the last stages of contracting wounds and regaining tissue function [95].

### 4.2. hVDAC1 Depletion Alters the Expression of ECM-Related Proteins and Stromal Factors

In cancer, dynamic and temporal alterations are involved in regulating adhesion, migration, proliferation, signaling, angiogenesis, EMT, and more. 

Our results show that in si-hVDAC1-2A-TTs, the expression of ECM-related genes was altered. ECM dynamics can result from changes in ECM composition such as altered synthesis or degradation of one or more ECM components or in its architecture because of altered organization. Our results show that VDAC1 depletion in cancer cells within the tumor altered the TME on several levels. The expression of most of the major ECM proteins, such as collagens, was significantly reduced in si-hVDAC1-2A-TTs (Figure 2A), as well as that of other components of the ECM, such as the glycoproteins, laminins, fibulins, fibrillins, fibronectins, and nidogens (Figure 3B). As ECM dynamics are indispensable for cancer progression involving ECM deposition and degradation, our results showing the reduced expression of the three major families of genes responsible for the breakdown of ECM constituents—MMPs, ADAMs and ADAMTSs (Figure 5A,B)—predict reduced ECM remodeling. This is further supported by the negative fold change of the LOX gene family levels (Figure 5C), indicating less deposition in the ECM.

Another important role of the ECM is to store and present stromal and growth factors, a role abused by cancer cells. A high level of stromal growth factors is usually found in the TME of various malignancies, and this helps to activate and subjugate the fibroblast population residing in the TME [96]. Our results showed a major decrease in TGF-β variants and their bioavailability in the ECM, as also reflected by a reduction in latent TGF-β-binding protein (LTBP) family members. TGF-β plays important roles in cancer and has tumor-promoting effects, increasing tumor invasiveness and metastasis [97].

One key target is the TGF-β, a multi-functional cytokine, that has been shown to prevent the growth and metastasis of certain types of cancers when inhibited [98]. The expression of another important growth factor, FAP, was reduced by si-hVDAC1-2A treatment. FAP expression has been observed in the activated stromal fibroblasts of more than 90% of all human carcinomas, with elevated levels of stromal FAP predicting a poor survival outcome [99]. 

Two key matricellular proteins, well studied for their roles in cancer progression, namely, tenascin C and periostin, were shown to be significantly reduced at both the mRNA and protein levels following hVDAC1 depletion (Figure 3D and Figure 4). Their expression levels correlated with tumor progression [100,101]. High expression of tenascin C in si-hVDAC1-2A-TTs occurred in the zone of tumor–stroma interaction (Figure 4). Periostin has been linked to EMT induction in cancer cells when overexpressed [102]. High serum levels of periostin or tenascin C were found to correlate with poor survival rates of patients (Figure 3E,F). While tenascin C expression is restricted to connective tissues and stem cell niches in adult tissues, it is very prominent in tumor tissues [103].

Other factors that were decreased in si-hVDAC1-2A-TTs are attributed to pro-angiogenic processes, namely, the PDGF and ANGPT families. The reason for increased expression of some pro-angiogenic factors, such as ENPP2 and PROX1 (Figure 6A), is not clear. We also observed an increase in some factors that inhibit angiogenesis, mainly CXCL10 and IL12. These two factors are also associated with the activation and recruitment of the immune system [104]. This suggestion is further supported by the increase in interferon gamma (IFNγ) (Figure 7B).

In summary, we found that depleting hVDAC1 in cancer cells leads to metabolic re-programming, tumor regression, and disruption of the tumor–host interactions. This is reflected in the alteration of a battery of genes associated with TME, including those involved in ECM organization, ECM structure, ECM-related peptidases and angiogenesis, as well as intercellular-interacting proteins, integrins, and growth factors associated with stromal activities (Figure 10). 

With the appearance of the TME as an essential element of cancer malignancy, therapies targeting the host compartment of tumors have begun to appear and are starting to be employed in the clinic [105]. This includes compounds targeting angiogenesis, with more than 1000 clinical trials with anti-angiogenic drugs, such as anti-VEGF antibodies and bevacizumab, having been conducted worldwide [105]. However, the overall survival benefits of anti-angiogenic drugs have been rather modest, with most cancer patients stopping responsiveness or not responding to such therapy [105]. The non-cellular compartment of the TME, such as the collagen cross-linking enzyme LOX and the metalloproteases MMPs, are also being considered as a target for cancer treatment. However, there are conflicting views on the use of LOX inhibitors in such treatment [72,106]. Clinical phase III studies of MMP inhibition in non-small-cell lung cancer were cancelled due to a lack of efficiency [94].

Finally, the link between cancer cell metabolism and the immune system is well established, as well as the ability of tumors to create an immunosuppressive microenvironment. The immune system affects the metabolic functions of the host organism, a phenomenon known as immunometabolism [107]. Moreover, the dysregulation of metabolites has been found in many pathologies, and the role of metabolite fluctuation caused by host immune cells versus pathogens is an active area of study [108].

Our results demonstrate increased immunogenicity as reflected in the increased expression of mouse MHC-class 1, MHC-class 2 genes, costimulatory molecules such as CD86, CD40, ICOS, CD274 (PD-L1—programmed cell death 1 ligand 1), and GITR, while molecules related to NK cells and cytotoxic activity such as granzyme and perforin were also increased (Alhozeel et al., unpublished data). We used nude mice that carry the FOXN1 mutation leading to an athymic phenotype lacking αβ-T cells that is defined as immunocompromised, lacking Th1, Th2, Th17, CD8+, and Treg cells. However, these mice also have cells of myeloid origin such as macrophages, granulocytes, antigen presenting cells (APCs), natural killer (NK) cells, B cells and T cells, such as gδ-T cells [109]. This mice phenotype demonstrates that anticancer immune surveillance potential is reduced in these mice, but not innate immunity and inflammatory activities, with the latter essential for tumor development. Yet, it is important to validate the effects of VDAC1 depletion on the immune system in a syngeneic mouse model and to compare the results obtained with node mice. Thus, the effects of VDAC1 silencing on the link between cancer cell metabolism and the immune system and the ability of tumors to create an immunosuppressive microenvironment are a topic for another study.

## 5. Conclusions

In conclusion, here, we have shown that a single treatment affecting metabolic rewiring by mitochondrial VDAC1 silencing attacked several components of the TME, ranging from structural proteins to MMPs and LOX, and elicited a stromal response similar to that seen in the reaction to a foreign body found in normal cancer stroma. Thus, we propose that attacking cancer cell metabolism via VDAC1 down expression reduces the cancer wound healing process associated with inflammatory and stromal activities related to the attenuation of the immune response. The results provide insights into the relationship between the TME and metabolism and point to VDAC1 as a novel target not only for reprogramming cancer cells, but also for modulating TME components.

## Figures and Tables

**Figure 1 cancers-13-02850-f001:**
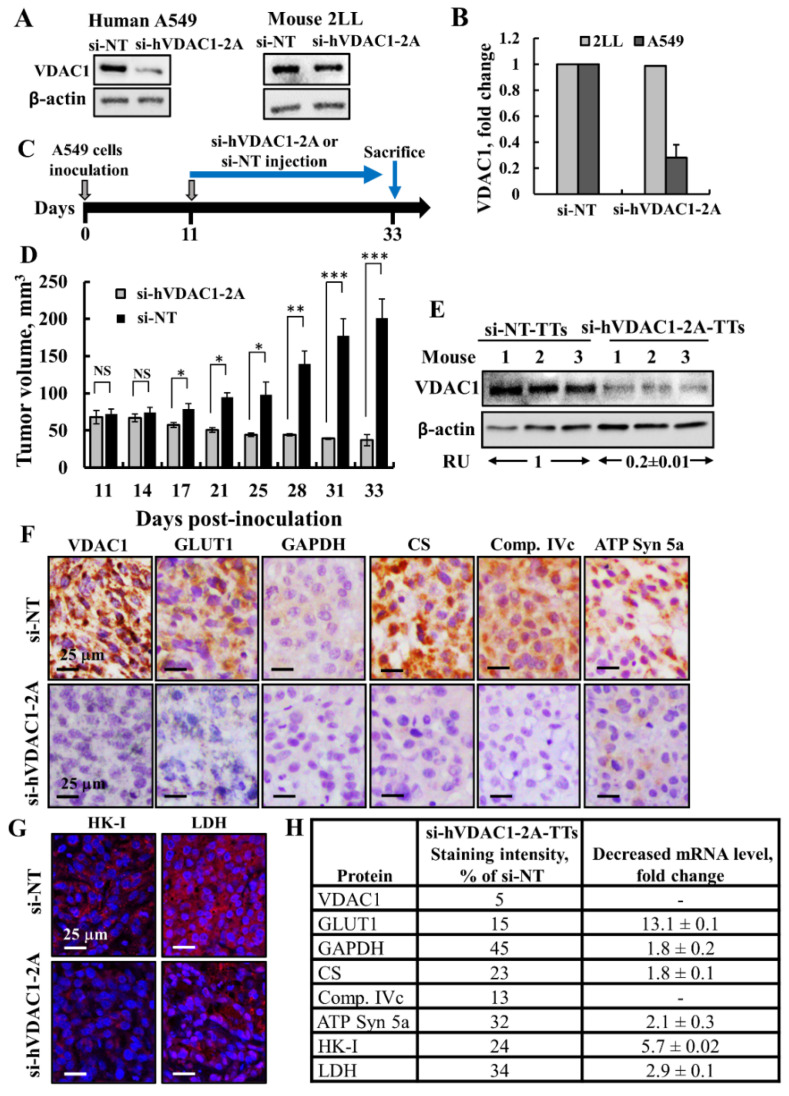
Human-specific siRNA si-hVDAC1-2A specifically silences human but not mouse VDAC1, inhibits lung cancer cell-derived xenograft growth, and reduces the expression of metabolism-related proteins. (**A**) A549 and 2LL cell lines were transfected with si-NT or si-hVDAC1-2A and analyzed for VDAC1 expression using anti-VDAC1 antibodies. Cells were transfected using jetPRIME reagent and 100 nM of siRNA (Appendix A for uncropped Western Blot). (**B**) Quantitative analysis of VDAC1 protein levels in the si-hVDAC1-2A-treated cells, relative to si-NT-treated cells, is presented as relative units (RUs). (**C**) Schematic presentation of the course of the experiment and siRNA treatment initiation. (**D**) A549 cells (3 × 10^6^) were subcutaneously (s.c.) inoculated into nude mice. When tumor size reached 70 mm^3^, the mice were divided into two matched groups, and xenografts were injected intratumorally every 3 days with si-NT (black bars, 7 mice) or si-hVDAC1-2A (gray bars, 8 mice) to a final concentration of 100 nM. The calculated average tumor volumes are presented as means ± SEM (* *p* ≤ 0.05, ** *p* ≤ 0.01, *** *p* ≤ 0.001), NS—non-specific. (**E**) si-NT-TT and si-hVDAC1-2A-TT sections from A549 xenograft mice were analyzed for VDAC1 levels by immunoblotting (Appendix A for uncropped Western Blot). Relative units (RUs) presented as the mean ± SEM; *n* = 3 mice. (**F**,**G**) Representative IHC staining using specific antibodies against VDAC1, GLUT1, GAPDH, citrate synthase (CS), complex IVc (Comp.IVc), and ATP synthase 5a (ATP Syn 5a) (**F**) or IF staining for HK-I and LDH (**G**) of si-NT-TT and si-hVDAC1-2A-TT sections derived from A549 xenografts. (**H**) Quantitative analysis of the levels of metabolism associated proteins IHC stained section intensity using Image J software, presented as the percentage of the si-hVDAC1-2A-TT staining intensity, relative to that in si-NT-TTs, and their levels as analyzed by qRT-PCR analysis of mRNA and presented as fold change.

**Figure 2 cancers-13-02850-f002:**
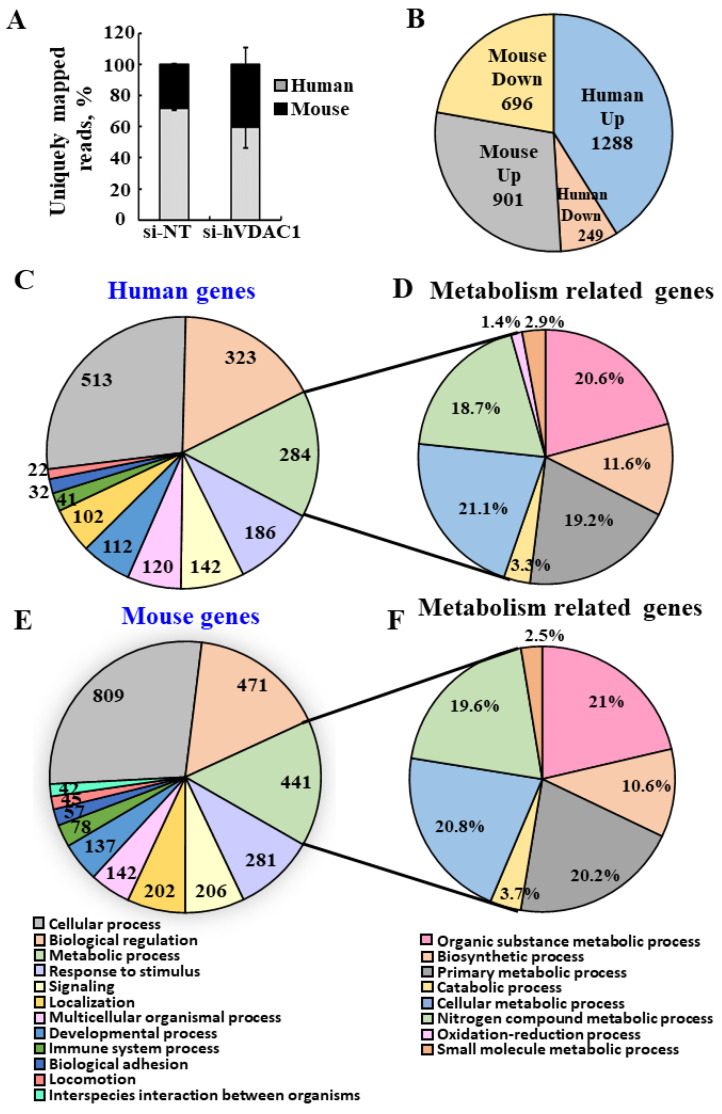
NGS gene analysis of mouse and human origin differentially expressed in si-hVDAC1-2A-TTs. RNA, isolated from tumors treated with si-hVDAC1-2A or si-NT (75 nM), was subjected to NGS with data subjected to bioinformatics analysis. (**A**) % of uniquely mapped reads for mouse and human. (**B**) No. of significantly up- and downregulated genes of human and mouse origin with a *p*-value < 0.05, and a linear fold change >1.5 or <−1.5 in si-hVDAC1-2A-TTs. (**C**–**F**) Classification of the human (**C**,**D**) and mouse (**E**,**F**) differentially expressed genes to biological processes, with the number of genes related to each process indicated inside the chart. Further breakdown of the metabolic processes to sub-classes is presented in (**E**,**F**) for human and mice, respectively. The analysis was carried out using the Panther gene list analysis, applying functional classification against GO-Slim Biological Process.

**Figure 3 cancers-13-02850-f003:**
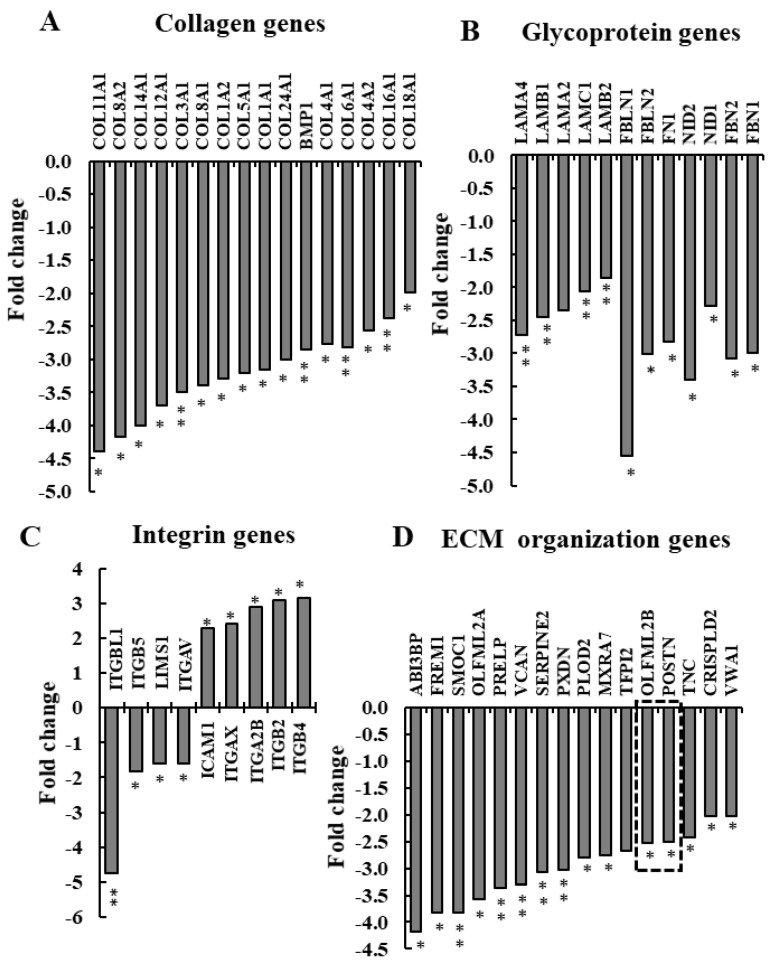
Differentially expressed ECM structure-related genes of murine origin. ECM structure-related genes found to be enriched using DAVID Gene Ontology analysis are presented as the fold change of the expression in si-hVDAC1-2A-TTs relative to si-NT-TTs. (**A**) Collagen genes. (**B**) Glycoprotein genes. (**C**) Integrin genes. (**D**) ECM organization genes. In all cases, * *p ≤* 0.05, ** *p ≤* 0.01. Two selected genes, periostin and tenascin C, whose expression was further analyzed at the protein level (Figure 3), are indicated by the dashed frames.

**Figure 4 cancers-13-02850-f004:**
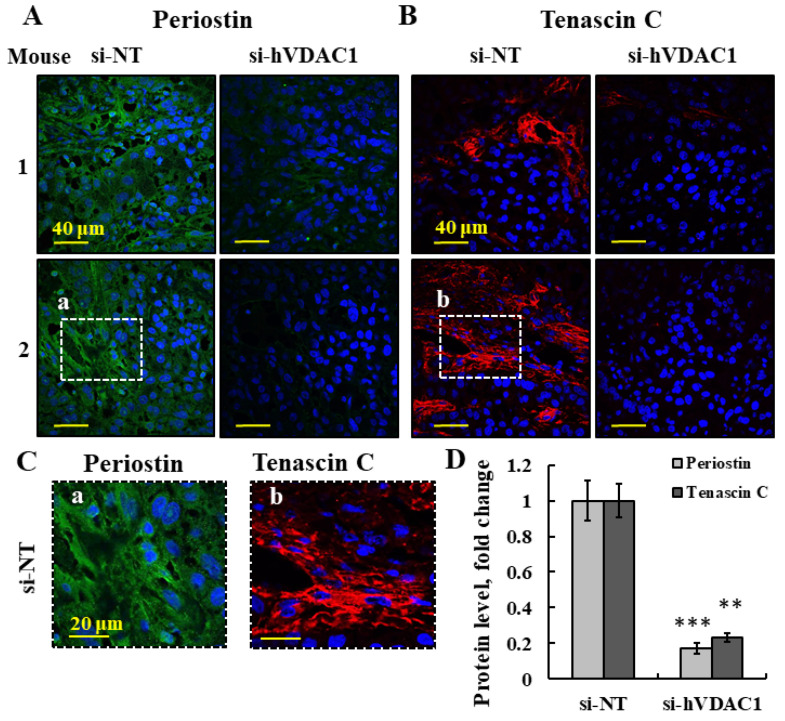
IF analysis of periostin and tenascin C expression in si-NT-TTs and si-hVDAC1-2A-TTs. (**A**,**B**) Representative IF staining using specific antibodies against periostin and tenascin C derived from A549 xenografts in si-NT-TT and si-hVDAC1-2A-TT sections. In (**C**), an enlargement of the area (**a**,**b**) in (**A**,**B**), respectively, is shown. (**D**) Quantitative analysis of fluorescence intensity in (**A**,**B**). The results are the means ± SD, ** *p ≤* 0.01, **** p ≤* 0.001. Furthermore, according to the TCGA database, lung cancer patients with elevated serum levels of periostin have a significantly lower overall survival rate than patients with low serum periostin levels (Appendix A). The same correlation was also found with serum tenascin C, where high levels in the serum were linked to poor overall survival probability (Appendix A).

**Figure 5 cancers-13-02850-f005:**
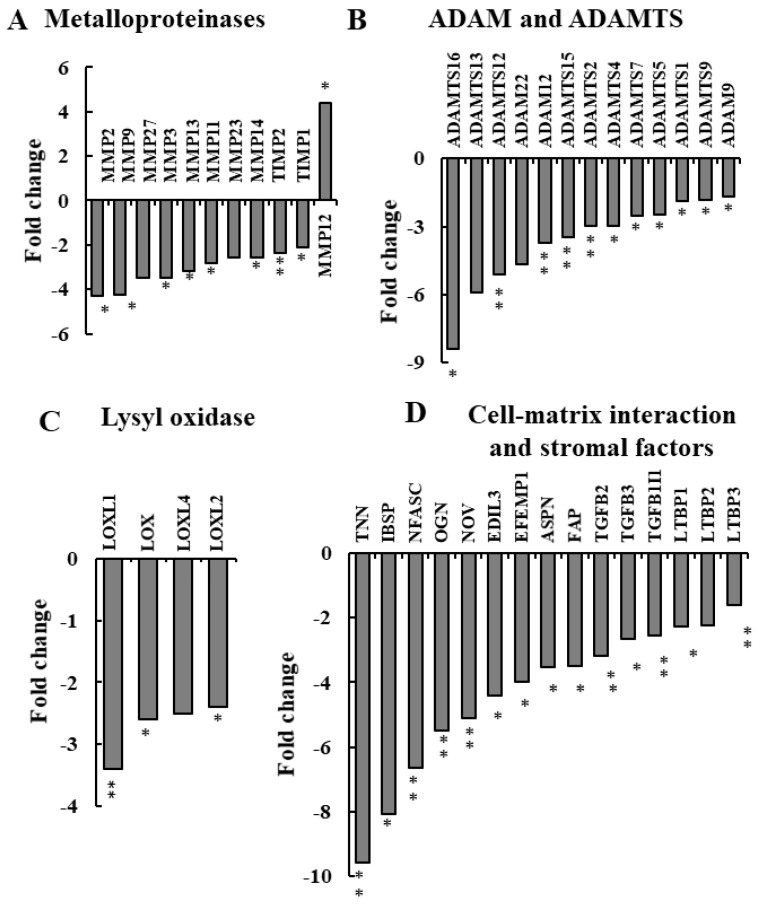
**Figure 5**. si-hVDAC1-2A reduces the expression of mouse genes involved in ECM deposition and degradation of TME factors. Differentially expressed ECM deposition and degradation genes of murine origin found to be enriched using DAVID Gene Ontology analysis are presented as the fold change in si-VDAC1-2A-TTs relative to si-NT-TTs. (**A**) Metalloproteinase genes. (**B**) ADAM and ADAMTS genes. (**C**) Lysyl oxidase and heparan sulphate genes. (**D**) Cell–matrix interactions- and stromal factor-related genes. In all cases, * *p ≤* 0.05, ** *p ≤* 0.01.

**Figure 6 cancers-13-02850-f006:**
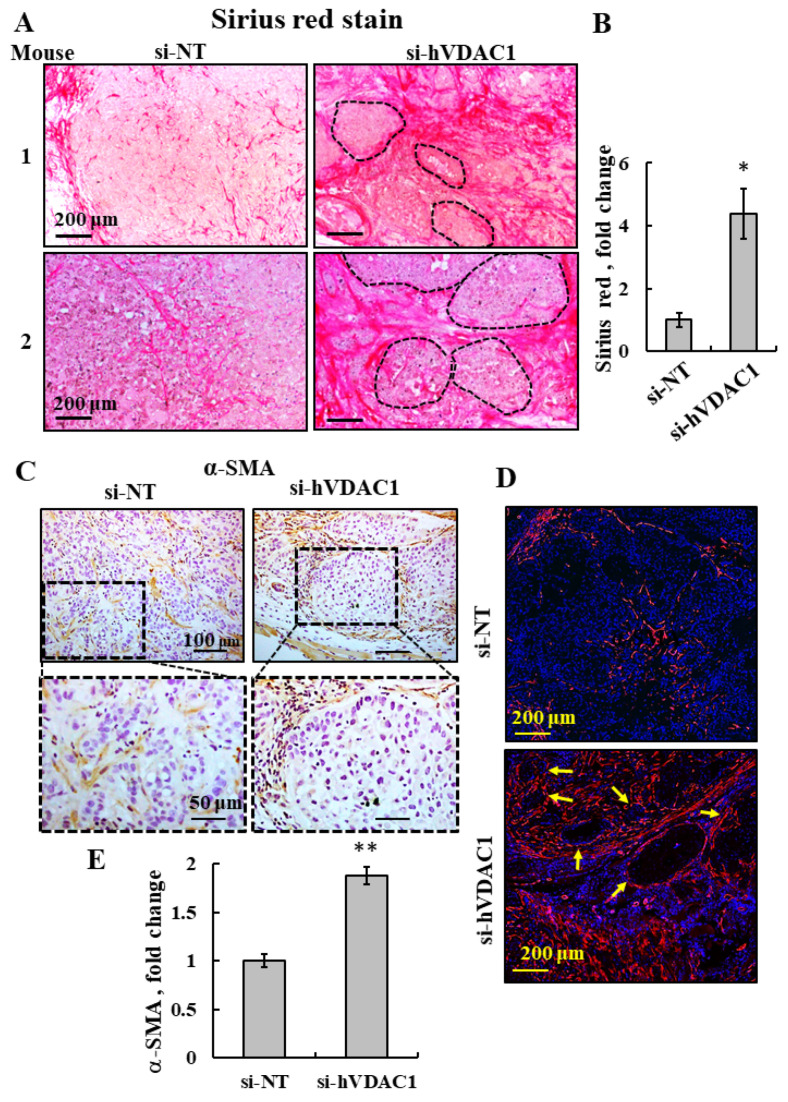
si-hVDAC1-2A-TTs show altered ECM organization and encapsulate cancer cells in the tumor. (**A**) Representative sections from two mice of si-NT-TT and si-hVDAC1-2A-TT sections derived from A549 xenografts stained with Sirius red. The dashed lines surround the encapsulated cancer cells within the residual tumor. (**B**) Quantitative analysis of Sirius red intensity. (**C**,**D**) Representative IHC (**C**) and IF (**D**) staining of si-NT-TT and si-hVDAC1-2A-TT sections with anti-α-SMA antibodies. Arrows point to the spherical organization of fibroblasts. (**E**) Quantitative analysis of α-SMA presented in (**D**). The results are the means ± SD, * *p ≤* 0.05, ** *p ≤* 0.01.

**Figure 7 cancers-13-02850-f007:**
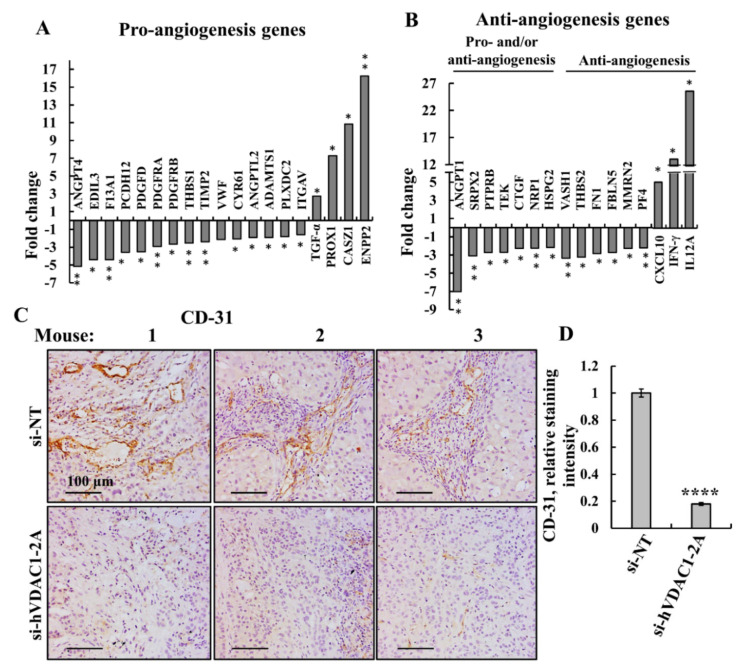
si-hVDAC1-2A tumor treatment alters the expression of angiogenesis-related mouse genes. Differentially expressed angiogenesis-related genes of murine origin found to be enriched using DAVID Gene Ontology analysis are presented as the fold change in si-VDAC1-2A-TTs relative to si-NT-TTs. (**A**) Pro-angiogenesis genes. (**B**) Anti-angiogenesis genes, as well as genes encoding for anti- or pro-angiogenesis factors, depending on the conditions. In all cases, * *p ≤* 0.05, ** *p* ≤ 0.01. Representative IF stained sections from si-NT-TTs and si-hVDAC1-2A-TTs derived from A549 xenografts stained with anti-CD-31 antibodies (**C**). Quantitative analysis of CD-31 staining intensity, presented as means ± SEM, **** *p* < 0.0001 (*n* = 3) (**D**).

**Figure 8 cancers-13-02850-f008:**
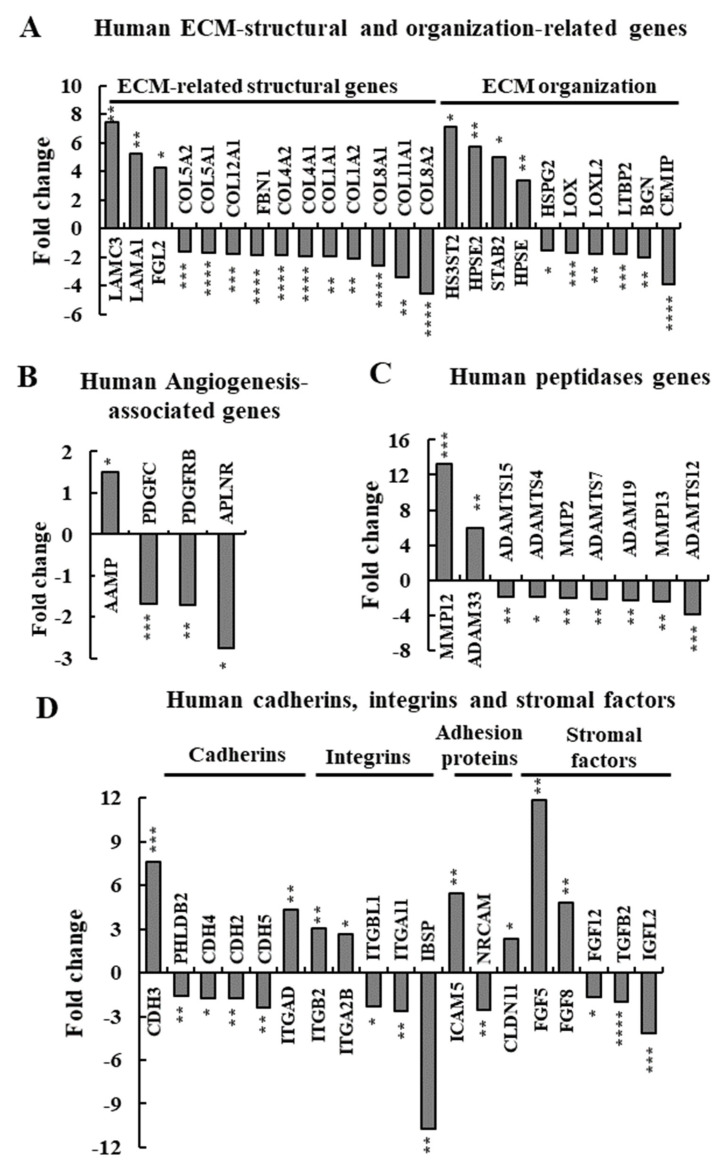
Altered expression of TME-related human genes in si-hVDAC1-2A-TTs. Differentially expressed genes of human origin found to be enriched using DAVID Gene Ontology analysis are presented as the fold change in si-VDAC1-2A-TTs relative to si-NT-TTs. (**A**) ECM structure- and organization-related genes. (**B**) Angiogenesis-associated genes. (**C**) Peptidase genes. (**D**) Cadherin, integrin, and stromal growth factor genes. In all cases, * *p* ≤ 0.05, ** *p* ≤ 0.01, **** p* ≤ 0.001, **** *p ≤* 0.0001.

**Figure 9 cancers-13-02850-f009:**
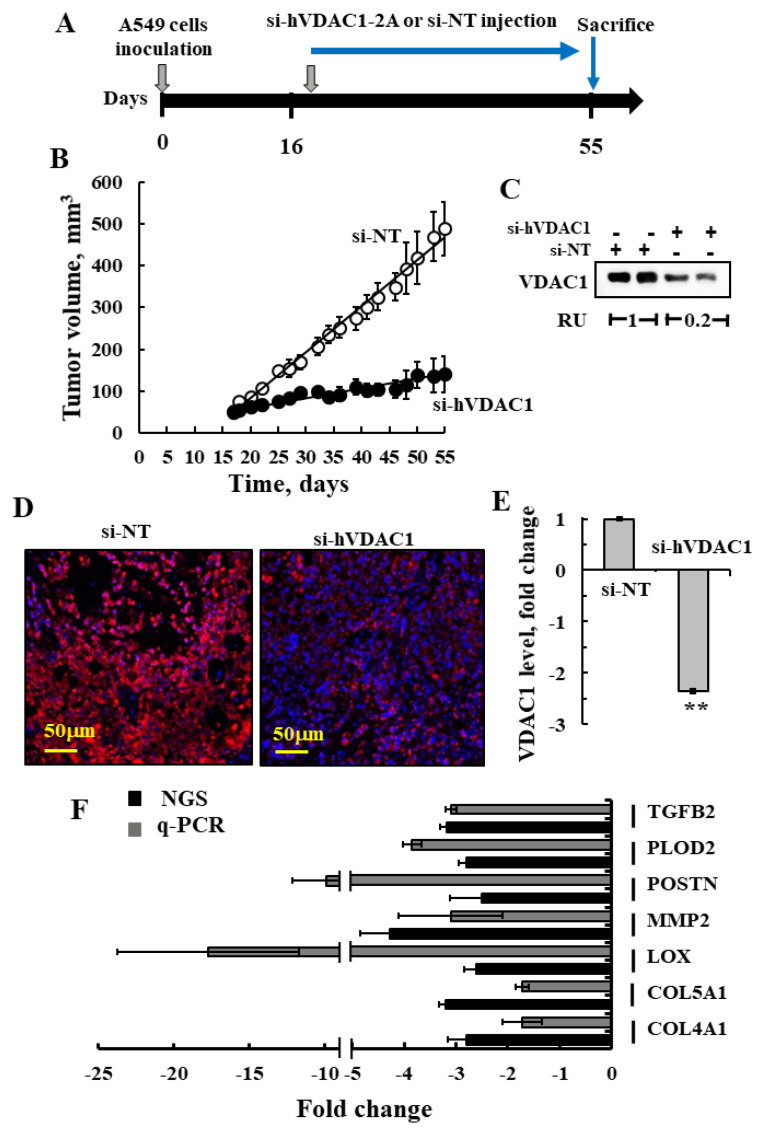
qRT-PCR validation of altered expression of selected TME-related genes. (**A**) Schematic presentation of the course of the experiment and peptide treatment initiation. (**B**) A549 cell-derived xenografts were established, and when tumor volume reached ~60 mm^3^, the mice were split into two groups and injected every 3 days with si-NT or si-hVDAC1-2A (200 nM), and tumor volume was measured. (**B**) Tumors in the si-NT-treated group reached an average of 490 mm^3^ at day 55 post-cell inoculation, while tumors in the si-hVDAC1-2A-treated group measured 140 mm^3^. Mice were sacrificed, and the tumors were excised and frozen in liquid nitrogen until used. (**C**) Tissue samples from excised tumors analyzed for VDAC1 levels by immunoblotting using anti-VDAC1 antibodies with the relative expression levels (RU) are presented in the blot (Appendix A for uncropped Western Blot). (**D**,**E**) Representative staining of si-NT-TT and si-hVDAC1-2A-TT sections with anti-VDAC1 antibodies with confocal images (**D**) and quantitative analysis of the fluorescence intensity (**E**) are shown. The results are the means ± SD, ** *p* ≤ 0.01. (**F**) The fold of change in the expression level of selected proteins was analyzed by q-PCR (gray bars), and data obtained from the NGS analysis (black bars) are presented. RNA isolation and qPCR of key TME genes using mouse-specific primers (Appendix A) were performed as described in Section 2.

**Figure 10 cancers-13-02850-f010:**
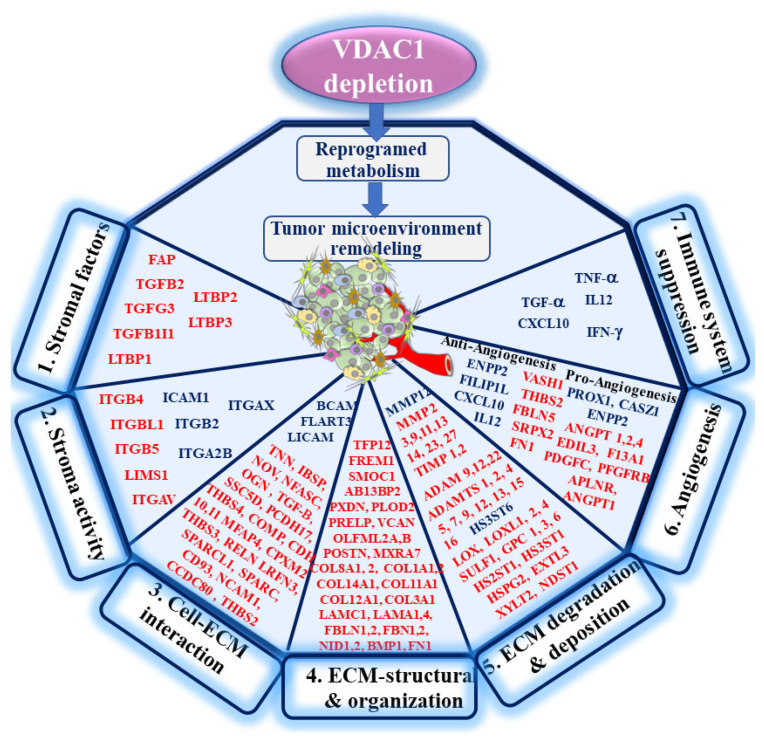
A schematic presentation of mitochondrial VDAC1 depletion and metabolic reprogramming leading to TME remodeling in A549-derived tumors. The overexpressed VDAC1 in mitochondria affects the homeostatic energy and metabolic states of cancer cells. Silencing hVDAC1 expression leads to a reprogramming of metabolism and to TME remodeling. This is reflected in alterations in the expression of genes related to: 1. stromal factors, 2. stroma activity, 3. cell–ECM interactions, 4. ECM structure and organization, 5. ECM degradation and deposition, 6. angiogenesis, and 7. immune system suppression. In each group, the major genes with enhanced expression (blue) or reduced expression (red) are presented. As tumor progression requires a cooperative interplay between host and cancer cells, and the ECM is intensively remodeled during cancer progression, VDAC1-induced alteration in the whole spectrum of TME-related genes affects both the tumor and reciprocal feedback between ECM molecules, host cells, and cancer cells. Tumors can influence the microenvironment by releasing extracellular signals, promoting tumor angiogenesis and inducing peripheral immune tolerance, while the tumor surrounding the microenvironment as immune cells can affect the growth and evolution of cancerous cells.

**Table 1 cancers-13-02850-t001:** List of chemicals and reagents used in this study.

Reagent	Catalog Number	Company
In vivo JetPEI-Transfection reagent	201-10G	PolyPlus (Illkirch, France)
JetPRIME- Transfection reagent	114-15	PolyPlus (Illkirch, France)
Triton X-100	T-6878	Sigma (St. Louis, MO, USA)
Tween-20	20452301	Bio-Lab ltd. (Jerusalem, Israel)
Paraformaldehyde (PFA)	15710	Emsdiasum (Hatfield, PA, USA)
Dulbecco’s modified Eagle’s medium (DMEM)	41965-039	Gibco (Grand Island, NY, USA)
Roswell Park Memorial Institute (RPMI) 1640	21875-034	Gibco (Grand Island, NY, USA)
Normal goat serum (NGS)	04-009-1A	Biological Industries (Beit Haemek, Israel)
Fetal bovine serum (FBS)	04-007-1A	Biological Industries (Beit Haemek, Israel)
L-glutamine	03-020-1C	Biological Industries (Beit Haemek, Israel)
Penicillin–streptomycin	03-031-5B	Biological Industries (Beit Haemek, Israel)
3,3-diaminobenzidine (DAB)	SK-4105	ImmPact-DAB (Burlingame, CA, USA)

## Data Availability

Data are contained within the article or Appendix A and the crude data are available on request.

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
