# Peer review of "VDAC1 Silencing in Cancer Cells Leads to Metabolic Reprogramming That Modulates Tumor Microenvironment"

_cancers, 2021, doi:10.3390/cancers13112850_

Round 1

Reviewer 1 Report

Broadly, the authors have shown the possible association of voltage-dependent anion channel 1 (VDAC1) with cellular metabolism and tumor microenvironment. VDAC1 is an abundant protein in the outer membrane of mitochondria and is known to be a gatekeeper of several metabolites. Herein authors studied that the knockdown of VDAC1 altered cellular metabolism and suppressed glucose transporter-1, citrate synthase, ATP synthase 5a, mitochondrial Complex IV etc. The authors performed NGS analysis on RNA, extracted from si-hVDAC1 treated Human lung cancer and mouse cancer cells xenograft and analyzed the differential distribution of proteins of key cellular and metabolic pathways, and also analyzed the change in expression of proteins involved in ECM, structural proteins, and angiogenesis. Furthermore, the authors have visualized Collagen fibers with Sirius red in si-hVDAC treated tumor. Though Authors have substantially addressed the concerns raised in the previous reviews. However, I still have few sustained concerns;

The authors performed NGS analysis of tumor xenograft, which was treated with intratumoral si-hVDAC1. How do the authors confirm the changes observed in cellular metabolism are due to the administration of si-hVDAC1 not an artifact of treatment? The authors haven't shown any evidence of whether VDAC1 knockdown affects the rate or, turnover of any metabolites? Since mitochondria is a major site of oxidation-reduction reaction and inhibition of its major protein VDAC1 could affect mitochondrial homeostasis. Though authors provided supporting evidence that VDAC1 knockdown marginally downregulated Comp IV, ATP syn 5A etc. However, there is no such evidence has been provided by authors, which can support whether VDAC1 silencing led to metabolic reprogramming. I would suggest authors to determine the metabolic flux analysis in these treatment conditions if possible? 

I can agree with the authors that VDAC1 silencing influence the microenvironment. Foxn1nu homozygous nude mice lack T cells and suffer from a lack of cell-mediated immunity, how can authors be sure that intratumoral administration of si-hVDAC exposed to human cancer cells alters the microenvironment in a mouse? I am wondering if the authors can elaborate their study design in detail and explain. Did the authors encapsulate the cancer cells with ECM? Authors need to provide the details of mice used in this study such as vendor ID, genotype, and complete tumor inoculation procedure in the method section.   

I would also suggest authors to consider rephrasing the title of the manuscript as it is not explicitly articulating the main text of the manuscript.   

Method; sections 2.3, 2.5 are quite inadequately written and need to briefly describe, and should be self-explanatory. 

Author Response

Reviewer 1

Broadly, the authors have shown the possible association of voltage-dependent anion channel 1 (VDAC1) with cellular metabolism and tumor microenvironment. VDAC1 is an abundant protein in the outer membrane of mitochondria and is known to be a gatekeeper of several metabolites. Herein authors studied that the knockdown of VDAC1 altered cellular metabolism and suppressed glucose transporter-1, citrate synthase, ATP synthase 5a, mitochondrial Complex IV etc. The authors performed NGS analysis on RNA, extracted from si-hVDAC1 treated Human lung cancer and mouse cancer cells xenograft and analyzed the differential distribution of proteins of key cellular and metabolic pathways, and also analyzed the change in expression of proteins involved in ECM, structural proteins, and angiogenesis. Furthermore, the authors have visualized Collagen fibers with Sirius red in si-hVDAC treated tumor. Though Authors have substantially addressed the concerns raised in the previous reviews. However, I still have few sustained concerns;

Comment 1. The authors performed NGS analysis of tumor xenograft, which was treated with intratumoral si-hVDAC1. How do the authors confirm the changes observed in cellular metabolism are due to the administration of si-hVDAC1 not an artifact of treatment?

 In this study, as in our other previous studies, we compare the effects of intratumoral-injected si-hVDAC1 to that of si-NT (non-targeting siRNA) given to a tumor in the same way, and in the same concentration as calculated according to the tumor volume. In a previous study (Arif et al., 2014, Ref 41, Fig. 8a–d, see below), we compared the growth of a tumor untreated and treated with si-NT and found that the si-NT slightly decreased the tumor growth relative to untreated tumors, but considerably less than si-hVDAC1.

Furthermore, the effects of si-VDAC1 were obtained when given i.v. in an intracranial GBM mouse model (Ref. 48). Thus, si-hVDAC1 could not be an artifact of treatment.

Comment 2- The authors haven't shown any evidence of whether VDAC1 knockdown affects the rate or, turnover of any metabolites?

Indeed, analysis of metabolites in the tumor is another way to look for changes in the cell metabolism. Our approach is to analyze the expression of metabolism-related enzymes in the tumor as a reflector of metabolic activity. In a xenograft mouse model, it is difficult to follow the rate or turnover of metabolites. First, it is necessary to have the use of a large number of mice for each time point—at least 5 mice and 4–5 time points. Second, this requires a different approach—the use GC/MS. Third, as one can see in published papers, analyzing metabolites needs the right expertise and the focus should only be on this.  We collaborated with an expert in the metabolite analysis field, and found many changes in the levels of metabolites, including in amino acids, lipids, and other less known compounds. Thus, this approach required a distinct study.

Comment 3- Since mitochondria is a major site of oxidation-reduction reaction and inhibition of its major protein VDAC1 could affect mitochondrial homeostasis. Though authors provided supporting evidence that VDAC1 knockdown marginally downregulated Comp IV, ATP syn 5A etc. However, there is no such evidence has been provided by authors, which can support whether VDAC1 silencing led to metabolic reprogramming. I would suggest authors to determine the metabolic flux analysis in these treatment conditions if possible? 

 We and others define metabolic reprogramming as a reduction in the metabolic activity of a cancer cell to a level similar to that of non-cancerous cells. Silencing VDAC1 leading to metabolism reprograming has been presented in several of our published papers using cell lines and different cancer mouse models (see Refs. 41,46–52). We showed that silencing VDAC1 alters the expression of about 2,000 genes, many of which belong to mitochondria, glycolysis, and other pathways associated with metabolism (also shown here in Fig. 2).  In a cell-based study, we a showed decrease in mitochondrial membrane potential and ATP levels and, as in tumors, a decrease in many glycolysis and oxidative phosphorylation enzymes.

As to metabolic flux analysis, my response to comment 2 is also valid here.

Comment 4 - I can agree with the authors that VDAC1 silencing influence the microenvironment. Foxn1nu homozygous nude mice lack T cells and suffer from a lack of cell-mediated immunity, how can authors be sure that intratumoral administration of si-hVDAC exposed to human cancer cells alters the microenvironment in a mouse? I am wondering if the authors can elaborate their study design in detail and explain. Did the authors encapsulate the cancer cells with ECM? Authors need to provide the details of mice used in this study such as vendor ID, genotype, and complete tumor inoculation procedure in the method section.   

As presented in the Introduction, each tumor is composed of cancer and host cells (here mouse of origin) defined as the tumor microenvironment. The tumor microenvironment is comprised of a broad array of stromal, endothelial, immune, and inflammatory cells. The malignant cells and cells within the tumor microenvironment (TME) continuously interact with each other to develop a dynamic and tumor-promoting milieu [see Hanahan D, Coussens LM (2012) Accessories to the crime: functions of cells recruited to the tumor microenvironment. Cancer Cell 21(3):309–322]. Thus, when we fixed the tumors, we have both the cancer cells together with their microenvironment cells and structures as collagen fibers. As indicated in the underlined sentence, there is a cross-talk between the cancer cells and the microenvironment components. 

Most importantly, as we used siRNA specific to human VDAC1, we are disrupting the metabolism homeostasis of the cancer cell, but not in the cells of the TME, thus, enabling separation of the bidirectional cross-talk between malignant cells and the TME.  Moreover, the NGS analysis allows changes in the genes of human or murine origin to be followed. 

Our results show an alteration in the expression of a battery of genes associated with the TME, including those involved in the extracellular matrix organization and structure, matrix-related peptidases, angiogenesis, intercellular-interacting proteins, integrins, and growth factors associated with stromal activities.

This is well reflected in all NGS-associated results (Figs. 3–9 and Tables S3–S13) and when staining with a-SMA, Periostin, and Tenascin C.  Here, we added  to Fig. & panel C and D showing CD-31 staining for imaging angiogenesis.

 As to: elaborate their study design in detail and explain

At the end of the Introduction, we presented the study design and rationale.

In addition, we added the following to the Supplementary Materials:

 Here, we present the complete obtained data that in the form of tables with selected representative results were presented in the main article in the form of figures. Briefly, the study was designed to address the relationship between tumor metabolism and the tumor microenvironment (TME), we used siRNA specifically with human VDAC1, as shown previously and here, that it disrupts cancer energy and metabolism homeostasis and induces metabolic reprogramming in the cancer cells derived from human cells. This allows us to follow how metabolic reprograming of cancerous affects the properties of non-cancerous cells (TME) within the tumor. To explore the interplay between metabolic reprograming of cancer cells and non-cancerous cells representing the TME within the tumor, we performed a next-generation sequencing (NGS) analysis of human A549 cell-derived tumors in a mouse model treated with non-targeting si-RNA (si-NT) or with siRNA specifically targeting human VDAC1 (si-hVDAC1-2A). As the NGS analysis allows for distinguishing between genes of human and mouse origin, we were able to demonstrate the tumor–host interactions in lung cancer.

We have added  to 2.4 the details of mice used in this study.

Comment 5 - I would also suggest authors to consider rephrasing the title of the manuscript as it is not explicitly articulating the main text of the manuscript.   

  The modified new title is: “VDAC1 silencing in cancer cells leads to metabolic reprogramming that modulates the tumor microenvironment”

Instead of the previous one:

 VDAC1 silencing in cancer cells leads to metabolic and tumor microenvironment reprogramming: Two sides of the same coin

Method; sections 2.3, 2.5 are quite inadequately written and need to briefly describe, and should be self-explanatory. 

 As indicated in Section 2.3 the method is present in the Supplementary Material section.

For 2.5, we added the following:

Sirius red staining for collagen was carried out as described previously (56).

Briefly, tumor tissues were fixed and embedded in paraffin, and sections were stained with a 0.1% sirius red-picric solution. Sections were washed rapidly with acetic acid and photographed under a light microscope (Leica DM2500).

Reviewer 2 Report

The authors analyse the effects of VDAC1 silencing in A549 cells (Lung Cancer cell) on metabolic changes in the cells and the tumoral microenvironment.

The study is interesting however several important issues need to be clarified.

  • The authors show an effect VDAC1 silencing in tumor cells on the microenvironment. Are the observed modifications due to VDAC1 depletion or the decrease of the tumor size? The tumor volume is significantly reduced upon VDAC1 silencing and this affects the dialogue between the tumor and the stroma. Indeed, because the tumors are smaller angiogenesis is affected as well as the density of the stroma thus interfering with cancer cell metabolism. In summary, the observed modifications may be independent of VDAC1 and could be observed with other intervention that leads to the inhibition of tumor growth. This is to my opinion a major issue that need to be solved.
  • The experiment is done in one cell line only A549, is it relevant to other model cell lines?
  • The authors mention in the title that they target mitochondrial VDAC1, is there another VDAC1? If yes, what is the specificity of the siRNA against the mitochondrial VDAC1?

SPECIFIC POINTS:

Figure 1: the quantification of the IHC is missing for HK-1 and LDH.

The number in the legend do not correspond to the right panel for E,F,G.

Figure 4: I do not find relevant to add the survival analysis in the main figure since these clinical data are not in relation with the VDAC1 status in the patients’ tumors.

Figure 7 and 8: the analysis of mRNA expression is interesting however; the protein expression is much more relevant and should be shown for some of them.

Minor point: line 340-341, the sentence has been cut.

Author Response

 The authors analyse the effects of VDAC1 silencing in A549 cells (Lung Cancer cell) on metabolic changes in the cells and the tumoral microenvironment.

The study is interesting however several important issues need to be clarified.

  • Comment 1- The authors show an effect VDAC1 silencing in tumor cells on the microenvironment. Are the observed modifications due to VDAC1 depletion or the decrease of the tumor size? The tumor volume is significantly reduced upon VDAC1 silencing and this affects the dialogue between the tumor and the stroma. Indeed, because the tumors are smaller angiogenesis is affected as well as the density of the stroma thus interfering with cancer cell metabolism. In summary, the observed modifications may be independent of VDAC1 and could be observed with other intervention that leads to the inhibition of tumor growth. This is to my opinion a major issue that need to be solved.

 It is reported that cancer cells without blood circulation grew to 1–2 mm3 in diameter and then stopped growing, and that tumors may become necrotic or even apoptotic. A vascular support and tumor microenvironment is developed when the tumor volume is 1–2 mm3  (Holmgren et al, 1995; Parangi et al 1996).

In our study, we started the treatment with the si-RNA when the tumor volume reached 60–70 mm3, which is about 30-fold larger, suggesting that the tumor microenvironment was already developed.

The analysis of the tumors at the end of the experiment when the tumor volume of the si-hVDAC1- treated tumor was reduced over 2-fold from its volume before the start of the treatment.

  • Comment 2- The experiment is done in one cell line only A549, is it relevant to other model cell lines?

In this study, we used A549 in a xenograft mouse model. However, similar results were carried out with U-87MG cells, a glioblastoma cell line, showing similar effects of si-VDAC1 on the tumors (Refs. 41,46–52). In Ref. 48 (Table S4), we showed alterations in the expression of metallopeptidases (MMP) and inhibitors of their transcription, extra-cellular matrix proteins (EMC), and integrin and angiogenesis and angiogenesis inhibitors. In addition, using q-RT-PCR, immunoblotting, IHC staining, or DNA microarray analysis, we showed the alteration in angiogenesis (Fig. S7).

We plan to do a similar study using a synergic mouse model.

Comment 3- The authors mention in the title that they target mitochondrial VDAC1, is there another VDAC1? If yes, what is the specificity of the siRNA against the mitochondrial VDAC1?

 We have added to the Introduction the following:

In mammals, three isoforms of VDAC (VDAC1, VDAC2, and VDAC3) have been identified, and they have been shown to share some, but not all, structural and functional properties. VDAC1 is the most abundant and best studied isoform and is highly expressed in different tumors, including lung cancer, pointing to its significance in high energy-demanding cancer cell (Ref. 42). We added this information to the section about VDAC isoforms and why VDAC1 was selected for this study.

As shown in Ref. 41, (Arif et al., 2014, Fig.  1b),  the siRNA suppressed the expression of VDAC1, but had no effect on VDAC2 and VDAC3.

This is why we defined it as specific to human VDAC1.

Moreover, expression of murine VDAC1 restored cell proliferation in the presence of  siVDAC1 (Fig. 1H in Ref. 48, Arif et al . Neuro Onc,2017).

SPECIFIC POINTS:

Figure 1: the quantification of the IHC is missing for HK-1 and LDH.

 As requested, the quantitative analysis of HK-1 and LDH staining intensity is now added to Fig. 1

The number in the legend do not correspond to the right panel for E,F,G.

 Thanks, this is now corrected

Figure 4: I do not find relevant to add the survival analysis in the main figure since these clinical data are not in relation with the VDAC1 status in the patients’ tumors.

 We have moved panels E and F to the Supplementary Data section as new Fig. S2.

Figure 7 and 8: the analysis of mRNA expression is interesting however; the protein expression is much more relevant and should be shown for some of them.

We clearly understand this reviewer’s request to demonstrate some of the proteins whose expression levels were analyzed by NGS to be analyzed also by immunoblotting or IF.  However, due to a freezer accident, we lost our samples. 

We would like to note that our results show good correlation between the mRNA and protein levels, for example, for collagens (Fig. 3A and Fig. 5A), and for periostin and tenascin C (Fig. 3D and Fig. 4).

We now perform for Fig. 7, immunofluorescent staining for CD-31 we stained an angiogenesis marker, reflecting changes in many factors contributing to the altered angiogenesis.

 Fig. 8 represents changes in human origin genes (implemented cancer cells).  The analysis of human origin proteins is complicated and is known as a major limitation in studies related to tumor–host interactions. It is difficult to separate cancerous from non-cancerous signaling pathways within a tumor. This is one of the advantages of this study, using siRNA specifically targeting human VDAC1 and NGS analysis allows us to distinguish between genes of human and mouse origin, enabling demonstration of the tumor–host interactions in lung cancer. 

Separation of the human origin cells  from the mouse cells in the tumor is very complicated, particularly from frozen tissue. In addition, since most antibodies recognize both human and mouse proteins, we have no way to identify the source of the proteins nor to provide for Fig. 8 confirmation of the NGS data.

Minor point: line 340-341, the sentence has been cut.

Thanks, this is now corrected

Round 2

Reviewer 2 Report

Eventhough the authors answered some of my questions, I'm not convinced by their conclusion, but the work was performed in a proper manner.

This manuscript is a resubmission of an earlier submission. The following is a list of the peer review reports and author responses from that submission.

Round 1

Reviewer 1 Report

This study, “Mitochondrial VDAC1 silencing leads to metabolic rewiring and tumor micro-environment reprogramming: Two sides of the same coin.” used Next-generation sequencing (NGS) analysis to show how siVDAC1 A549 tumors altered mouse and human gene expression. The author concluded that “depleting VDAC1 in cancer cells led to metabolic reprogramming, tumor regression, and disruption of tumor-host interactions.” This conclusion is not fully supported by the results, especially for the metabolic reprogramming part. The authors only show the repressed metabolic genes by staining in Figure 1, but all the other figures are only about the tumor microenvironment (TME), which is also summarized in Figure 9. Although VDAC1 is a mitochondrial protein, the author can’t assume the effect of VDAC1 is all through metabolic alterations. As the current writing is a little misleading, metabolic rewiring should be removed from the title and tuned down in the abstract. This manuscript could be an interesting study of TME, after carefully re-writing.   

Figures 2, 3, and 5 can be re-arranged. Legend of figure 2 showed that “Two selected genes, periostin and 356 tenascin C, whose expression was further analyzed at the protein level (Fig. 3)” The flow will be better if 2 and 3 would be combined. Fig 5 could be combined with Fig 2A, B, as Fig 5 shows the confirmation of collagens alteration in Fig 2. 

Figure 7B didn’t show CXCL10, as discussed on line 567.

Author Response

Comment 1. The author concluded that “depleting VDAC1 in cancer cells led to metabolic reprogramming, tumor regression, and disruption of tumor-host interactions.” This conclusion is not fully supported by the results, especially for the metabolic reprogramming part.

We would like to indicate that the suggestion that silencing VDAC1 expression inducing metabolic reprograming is based on our recent 9 publications (see the list below) presenting the reversal of the metabolic pathways similar to that in non-cancerous cells, using different methods such as IF, IHC, q-PCR, proteomics, and NGS, and in several cancer types. Moreover, VDAC depletion also affects epigenetics as a mitochondrial metabolite-dependent process (Refs. 8, 9 below). Thus, in this paper, we focused on the effect of metabolism modification on the tumor microenvironment.

In Fig. 1 E and 1F, we presented representative results to show that indeed in the specific experiment used for NGS, the same major changes in the expression of metabolism-related enzymes represent glucose transport, glycolysis, the TCA cycle, electron transport, and ATP synthesis.

We now added metabolism-related enzymes analyzed using q-PCR results (Fig. 1G). In addition, we added NGS results as new Figure 2 to present the cellular functions altered upon VDAC1 silencing  and  to also show the changes in the expression metabolism-related genes.

Thus, we believe silencing VDAC1 resulted in reprogramming the cancer cell metabolism.

We modified the Introduction section to emphasize the link between VDAC1 depletion and reprogramming of cancer cell metabolism in both cells in culture and in vivo in several cancer mouse models.

 As to metabolic rewiring should be removed from the title and tuned down in the abstract

I hope this reviewer considers our changes in the Introduction and the addition of the results related to metabolism as revealed using q-PCR (Fig. 1E) along with the results of NGS pathways analysis (new Fig. 2 C–F) to accept our view that indeed cancer cell metabolism is reprogrammed upon VDAC1 depletion.

It important to note that in this MS, we show the cross-talk between the cancer cells (humans origin) and host cells (mouse, microenvironment) by depleting specifically VDAC1 in the cancer cells to alter metabolism, thus, modulating the tumor microenvironment. This is why we submitted the MS to the special issue of Cancer "Tumor and Metabolism”.

We changed the previous title:” Mitochondrial VDAC1 silencing leads to metabolic rewiring and tumor microenvironment reprogramming: Two sides of the same coin to the following: “Mitochondrial VDAC1 silencing in cancer cells leads to metabolic changes and tumor microenvironment reprogramming: Two sides of the same coin

  1. Abu-Hamad, S. Sivan and Shoshan-Barmatz, V. (2006) The expression level of the voltage-dependent anion channel controls life and death of the cell. Proc. Nat. Ac. Sci. USA, 103(15): 5787-5792
  2. Arif, T, Vasilkovsky, L., Refaeli, Y. Konson, A. and Shoshan-Barmatz, V. (2014) Silencing VDAC1 expression by siRNA inhibits cancer cell proliferation and tumor growth in vivo, Molecular Therapy–Nucleic Acids 3: e159
  3. Arif, T., Krelin, Y. and Shoshan-Barmatz, V. (2016) Reducing VDAC1 expression induces a non-apoptotic role for pro-apoptotic proteins in cancer cells differentiation, Biophys. Acta. 1857(8):1228-42
  4. T., Krelin, Y, Nakdimon, I., Benharroch, D., Paul A. and Shoshan-Barmatz, V. (2017). VDAC1 is a molecular target in glioblastoma, with its depletion leading to reprogramed metabolism and reversed oncogenic properties, Neuro Oncol. 1;19(7):951-964.
  5. Arif, T. Paul, A. Krelin Y., Shteinfer-Kuzmine A and Shoshan-Barmatz V. (2018). Mitochondrial VDAC1 silencing leads to metabolic rewiring and the reprograming of tumor cells to advanced differentiated states. Cancers, 8;10(12). pii: E499.
  6. Arif, T. Amsalem Z. and Shoshan-Barmatz. (2019). Metabolic reprograming via silencing of mitochondrial VDAC1 expression encourages differentiation of cancer cells. Molecular Therapy Nucleic Acids. 17:24-37.
  7. Stein, O. Arif, T., Pittalas, S. Chalifa-Caspi, V. and Shoshan-Barmatz, V. (2019) Rewiring of cancer cell metabolism by VDAC1 depletion resulted in time dependent manner tumor reprogramming: Glioblastoma as a concept, Cells 8(11). pii: E1330
  8. Amsalem , Arif T., Shteinfer-Kuzmine A., Chalifa-Caspi, V. and Shoshan-Barmatz, V. (2020) The mitochondrial protein VDAC1 at the crossroads of cancer cell metabolism: The epigenetic link, Cancers, Apr 22;12(4):1031
  9. Shteinfer-Kuzmine, A., Verma, A., Arif, T., Aizenberg, O., Paul A., and Shoshan-Barmatz (2020) Mitochondria and nucleus cross-talk: signaling in metabolism, apoptosis and differentiation and function in cancer IUBMB Life, October 2020 DOI: 10.1002

Comment 2- Figures 2, 3, and 5 can be re-arranged.

Legend of figure 2 showed that “Two selected genes, periostin and tenascin C, whose expression was further analyzed at the protein level (Fig. 3)” The flow will be better if 2 and 3 would be combined.

This is good idea, however, by doing this, the various panels in the figure will become too small to clearly see the results. Therefore, the results related to periostin and tenascin C were presented in the last panel (D) in Fig. 3 (formerly Fig. 2), and are immediately followed by Fig. 4 focusing on periostin and tenascin C.

Fig 5 could be combined with Fig 2A, B,

In Fig. 2A and 2B (now Fig. 3A and 3B), we present the changes in collagen and glycoprotein expression levels. However, in Fig. 5 (now Fig. 6), we stained the tumor sections with Serious red and the alpha smooth muscle actin, a-SMA, in order to visualize the re-organization of the ECM. The ECM is a dynamic component, involved in protein production and degradation, attributed to the expression levels of matrix metallo-proteinases, collagens producing fibroblasts, and other proteins that were presented in Figs. 4 and 5 (formerly Figs. 3 and 4). Thus, by combining Fig. 6 with 3A and 3B, we focus on collagens, but not on other aspects of ECM organization as also revealed by staining for fibroblasts using α-SMA as a marker to show that the fibroblasts encapsulate the tumor cells in VDAC1-deplated residual tumors, in contrast to the random fibroblast localization in VDAC1-expressing tumors.

Reviewer 2 Report

The current research article entitled,"Mitochondrial VDAC1 silencing leads to metabolic rewiring and tumor micro-environment reprogramming: Two sides of the same coin" is quite interesting and has scientific merits to be considered for publication. Authors have delivered intratumoral siRNA -human VDAC1 in A549 lung cancer xenograft tumor in nude mice and determined the tumor growth and other regulatory protein of glucose and mitochondrial homeostasis. Further, the authors determined angiogenesis markers and extracellular matrix protein expression change in siRNA-hVDAC1 treated xenograft mice. The study designed is clear, however, the specificity of siRNA-hVDAC1 and its delivery was published in the author's previous publication (Ref. 42-49). Herein, the authors haven't shown any immune cell markers, which is a major component of TME and has a downside of the manuscript to consider "tumor micro-environment reprogramming". The authors haven't included any internal positive control to compare the anti-tumor efficacy of siRNA-hVDAC1 in the current study? 
I believe the introduction is quite lengthy and can be improved with appropriate citations and which can be further described in the discussion. 
Method; immunoblotting; provide the catalog number of VDAC1 ab reactive to human and mouse and also provide the details of other antibodies used in the study. 

Immunohistochemistry (IHC); I am wondering if the authors performed parallel staining with antigen control and/or isotype IgG control? The authors need to provide IgG isotype control and normalize nonspecific binding of antibodies? 
Results: in figure1 footnote; the authors indicated that when tumor size reached 70 mm3  then the mice were divided, whereas figure 8 indicating 60 mm3 at the treatment start point. why did the authors use two different tumor sizes or it is a just typo error? Authors can depict steps of the tumor inoculation and, treatment regime, and harvesting in a figure.

Author Response

Comment 1

The authors haven't shown any immune cell markers, which is a major component of TME and has a downside of the manuscript to consider "tumor micro-environment reprogramming".

This reviewer is correct in that the microenvironment also includes immune cells. Indeed, we obtained changes in the expression of many immune system-related markers.  We used nude mice that carry the FOXN1 mutation, leading to an athymic phenotype lacking αβ-T cells, that is, defined, as immunocompromised lacking Th1, Th2, Th17, CD8+ and Treg cells. Thus, we thought these results should be presented together with a similar experiment carried out in a sub-cutaneous syngeneic mouse model.

Our results demonstrate accelerating increased immunogenicity as reflected in increased expression of mouse MHC-class1, MHC-class 2 genes, costimulatory molecules such as CD86, CD40, ICOS, CD274 (PD-L1- programmed cell death 1 ligand 1), GITR while molecules related to NK cells and cytotoxic activity such as granzymes and perforin were also increased.

We added the following text:

Our results demonstrate increased immunogenicity as reflected in the increased expression of mouse MHC-class1, MHC-class 2 genes, costimulatory molecules such as CD86, CD40, ICOS, CD274 (PD-L1- programmed cell death 1 ligand 1), and GITR while molecules related to NK cells and cytotoxic activity such as granzyme and perforin were also increased (Alhozeel et al., unpublished data). We used nude mice that carry the FOXN1 mutation leading to an athymic phenotype lacking αβ-T cells that is defined as immunocompromised lacking Th1, Th2, Th17, CD8+, and Treg cells.  However, these mice also have cells of myeloid origin such as macrophages, granulocytes, antigen presenting cells (APCs), natural killers (NK) cells, B cells and T cells, as NK T and gδ-T cells [109]. This mice phenotype demonstrates that anticancer immunesurveilance potential is reduced in these mice, but not innate immunity and inflammatory activities, with the latter essential for tumor development. Yet it is important to validate the effects of VDAC1 depletion on the immune system in a syngeneic mouse model and to compare the results obtained with node mice. Thus, the effects of VDAC1 silencing on the link between cancer cell metabolism and the immune system and the ability of tumors to create an immunosuppressive microenvironment are a topic for another study.

 Comment 2:

The authors haven't included any internal positive control to compare the anti-tumor efficacy of siRNA-hVDAC1 in the current study.

As we demonstrated, si-RNA targeted to VDAC1 has multiple effects including reprogrammed cancer cell metabolism, reversing oncogenic properties including reduced tumor growth, invasivity, stemness, EMT, angiogenesis, and tumor-associated macrophage presence, it is difficult to select a drug or treatment as a positive control. 

 Comment 3:

I believe the introduction is quite lengthy and can be improved with appropriate citations and which can be further described in the discussion

We modified the Introduction to address Reviewer 1 and this reviewer’s suggestions

Comment 4:

Provide the catalog number of VDAC1 ab reactive to human and mouse and also provide the details of other antibodies used in the study. 

We included in the Supplementary data a table (Table S1) presenting all antibodies used, their source, and dilutions used for immunoblotting, immunofluorescence, and immunohistochemistry.

 Comment 5:

 Immunohistochemistry (IHC); I am wondering if the authors performed parallel staining with antigen control and/or isotype IgG control? The authors need to provide IgG isotype control and normalize nonspecific binding of antibodies? 

In IHC staining, we selected the used antibodies based on their previous use in many published studies, showing specificity and the expected staining with respect to sub-cellular localization and that also were used in different tissues or tumor types.  Thus, we do not use new antibodies that need to be characterized.

 Comment 6

 In figure1 footnote; the authors indicated that when tumor size reached 70 mm3 then the mice were divided, whereas figure 8 indicating 60 mm3 at the treatment start point. why did the authors use two different tumor sizes or it is a just typo error?

The experiments in Fig. 1 and Fig. 8 are two different experiments; therefore, we presented the tumor growth time course in both figures. In Fig. 8 (now 9), we started the treatment with siRNA when tumor volume reached 60 mm3. This is indicated in the text in line 577: “Remarkably, although another A549-derived xenograft experiment treated with si-hVDAC1-2A A (Fig.  8A, B, now Fig. 9)”

As suggested, we have depicted steps of the tumor inoculation and treatment regime and harvesting in a panel added to Figs. 1 and 6.

1.            Harris, B.F., HE; Berry, ML; Bergstrom, DE; Bronson RT; Donahue, LR. , Nude 2 Jackson: A new spontaneous mutation in Foxn1 MGI Direct Data Submission: 2013.

Round 2

Reviewer 1 Report

The author didn't address my question that there is a strong link between VDAC1-regulated metabolic alteration and other TME changes. Although I give credit to the authors' previous study on metabolic role of VDCA1, I still feel this manuscript is more about TME, lacking the metabolic connections.